# Computational ranking identifies Plexin-B2 in circulating tumor cell clustering with monocytes in breast cancer metastasis

Emma Schuster[1,2,12], Nurmaa K. Dashzeveg[1,12], Fangjia Tong[1,12], Yuzhi Jia[1], Lamiaa El-Shennawy[1], Tong Zhang[3], Andrew D. Hoffman[1,4], Reta Birhanu Kitata[3], Golam Kibria[1], Youbin Zhang[5], Joshua R. Squires[1], Chunlei Zheng[6], Erika Ramos[1], Rokana Taftaf[1], David Scholten[1,2], Hannah F. Almubarak[1,2], Valery Adorno-Cruz[1], David P. Sullivan[7], Carolina Reduzzi[8], Allegra C. Minor[2,9], William Purev-Ochir[1], Sabina Spahija[5], Rong Xu[6], Kalliopi P. Siziopikou[7,9], Leonidas C. Platanias[5,9], Ami Shah[5,9], William A. Muller[7,9], William J. Gradishar[5,9], Massimo Cristofanilli[8], Chia-Feng Tsai[3,13], Tujin Shi[3,13] ✉ & Huiping Liu[1,5,9,10,11,13] ✉

Multicellular circulating tumor cell (CTC) clusters can be up to 50 times more efficient than single CTCs in mediating viable metastasis. Here, combining computational ranking and functional determination, we identify the trans-membrane protein Plexin-B2 (PLXNB2) as one of the top molecular targets associated with unfavorable distant metastasis-free survival, showing enriched expression in CTC clusters versus single CTCs from patients with advanced breast cancer (mostly female). Loss of PLXNB2 (Plxnb2) reduces the formation of homotypic tumor cell clusters and heterotypic tumor-myeloid cell clusters, reducing spontaneous metastases in female mice bearing human (mouse) breast cancer. Interactions of PLXNB2 with its ligands SEMA4C on tumor cells and SEMA4A on myeloid cells (monocytes) promote homotypic and hetero-typic CTC cluster formation, respectively, thereby driving lung metastasis. Global proteomic analysis reveals downstream effectors of the PLXNB2 path-way associated with tumor cell clustering. Thus, PLXNB2 is a therapeutic target for preventing new metastasis in breast cancer.

The rapid development of cutting-edge multi-omic analyses and computational modeling have facilitated the integration and trans-formation of bioinformatic data into phenotype-related discoveries[1–5]. Reciprocally, comprehensive experimental determination feeds back computational analysis-based data mining and prediction. While genomic and transcriptomic studies have been relatively extensive, our study seeks to synergize the power of phenotype-driving pro-teomic analysis and functional exploration for cancer discoveries. One

[1]Department of Pharmacology, Northwestern University Feinberg School of Medicine, Chicago, IL, USA. [2]Driskill Graduate Program in the Life Sciences, Northwestern University Feinberg School of Medicine, Chicago, IL, USA. [3]Biological Sciences Division, Pacific Northwest National Laboratory, Richland, WA, USA. [4]ExoMira Medicine Inc, Chicago, IL, USA. [5]Division of Hematology and Oncology, Department of Medicine, Northwestern University Feinberg School of Medicine, Chicago, IL, USA. [6]Center for Artificial Intelligence in Drug Discovery, Case Western Reserve University, Cleveland, OH, USA. [7]Department of Pathology, Northwestern University Feinberg School of Medicine, Chicago, IL, USA. [8]Division of Hematology and Medical Oncology, Department of Medicine, Weill Cornell School of Medicine, New York, NY, USA. [9]Department of Biochemistry and Molecular Genetics, Northwestern University Feinberg School of Medicine, Chicago, IL, USA. [10]Robert H. Lurie Comprehensive Cancer Center, Northwestern University Feinberg School of Medicine, Chicago, IL, USA. [11]Chan Zuckerberg Biohub Chicago, Chicago, IL, USA. [12]These authors contributed equally: Emma Schuster, Nurmaa K. Dashzeveg, Fangjia Tong. [13]These authors jointly supervised this work: Chia-Feng Tsai, Tujin Shi, Huiping Liu. ✉e-mail: Tujin.Shi@pnnl.gov; huiping.liu@northwestern.edu

of the most devastating features of solid tumors is distant spreading or stage IV cancer metastasis, predicting unfavorable overall survival for all breast cancers[6], especially estrogen receptor (ER) negative breast cancers, such as triple-negative breast cancer (TNBC) which lacks expression of ER, progesterone receptor, and epidermal growth factor receptor 2 (HER2)[7–9] with the lowest 5-year survival rate (~10%) after metastasis to the lungs, brain, and liver[10], followed by HER2-positive breast cancer (~40%)[6]. As such, our goal is to develop a computational ranking of protein candidates and identify new therapeutic targets that drive breast cancer metastasis.

Cancer is disseminated by circulating tumor cells (CTCs) that shed off the primary tumor and are capable of seeding and regenerating tumors in distant organs, including lung, liver, brain, and bone, due to their inherent and acquired properties, such as plasticity, proliferation, and intercellular interactions[11–14]. The dogma of single CTC-mediated cancer dissemination has been challenged by the detection of rare CTC clusters in the blood of patients with advanced breast cancer[11,15–19]. The existence of CTC clusters predicts unfavorable outcomes[11,15–19], as they are 20–50 times more likely to seed metastases than single CTCs[11,20–24]. To control and prevent metastatic disease, it is imperative to discover the diverse mechanisms of CTC clusters in cancer. Two cellular mechanisms have been proposed for CTC cluster formation[18]; one is collective dissemination or cohesive shedding[11,25], and another is tumor cell aggregation[16,26], which would lead to both homotypic and heterotypic tumor clusters, thereby promoting metastasis with unfavorable overall survival (OS)[18]. It remains an open question whether primary tumor proteomic profiles can be used to identify phenotypic drivers for early intervention, and guide targeting approaches to prevent homotypic and heterotypic tumor cluster formation and block metastasis.

In this work, we hypothesize that aggregation or cohesion phenotype-related adhesion proteins can regulate CTC cluster formation. Taking advantage of systems biology and mass spectrometry (MS)-based global proteomic data, we have developed a ranking method to assess all 608 cell adhesion molecule candidates (derived from the Molecular Signature Database[27–29] of the Gene Ontology Biological Processes[30,31]) in breast cancer. As a proof-of-concept, our study reveals a single-pass transmembrane protein, Plexin-B2 (PLXNB2), as one of the top candidates. Plexin-B2 is up-regulated in primary tumors and enhances the formation of both homotypic tumor cell clusters and heterotypic CTC-myeloid cell clusters in metastasis of breast cancer, especially TNBC.

## Results

### Proteomic ranking of cell adhesion molecules in breast cancer

To seek an unbiased in silico screen of candidate proteins in primary tumors that may regulate cancer progression and metastasis, we developed a mathematical ranking score, Rscore, to assess the significance ranking of adhesion network proteins in contributing to intercellular interactions. In our initial proof-of-concept analyses, the Rscore integrates individual rankings ($r_i$) of each adhesion molecule in various datasets with an adjustable weight of constant factors ($c_i$), including relative protein abundance (normalized intensity), tumor-specific expression versus normal adjacent tissues (p value, fold change, and absolute change), and clinical associations with OS and distant metastasis-free survival (DMFS) (p value and hazard ratio) (Fig. 1a, and Supplementary Fig. S1). We have assessed the Rscore rankings of the proteins, especially adhesion proteins overlappingly detected in lab-obtained and public datasets via MS proteome quantifications, including human breast tumors[32] (N = 122), TNBC tissue voxels versus normal adjacent regions (N = 6), breast cancer cell lines (N = 2), and patient-derived CTC specimens (N = 19), in association with clinical outcomes, especially OS and DMFS in multiple datasets of breast cancer.

Based on the MS proteomic analyses of human TNBC tissues from laser capture microdissection (tumor regions versus normal adjacent), we obtained a list of tumor-specific proteins, including 627 differentially expressed proteins in tumor tissues (Supplementary Data 1). After expanding the analysis to the tandem MS proteomic datasets of treatment-naive human breast tumors[32] (N = 122), 398 adhesion proteins were detected and ranked based on the relative protein abundance, calculated as spectral counts per protein and normalized by the ratio of a given protein's length (# of amino acids) versus the median protein length (Supplementary Fig. S1a, Supplementary Data 1). To further narrow down to the intrinsic adhesion proteins in cancer cells, we combined the MS proteomic profiles of human TNBC cells (MDA-MB-231[33] and Hs578T[34]) and identified 124 overlapped adhesion proteins with ranking in normalized peptide-spectrum match (PSM) counts (Supplementary Fig. S1a, and Supplementary Data 1-tabs 3–4). We also detected 258 adhesion proteins from patient CTCs and CTC-derived PDX CTC-205 and CTC-92 (N = 19 specimens) (Supplementary Fig. S1a, and Supplementary Data 1-tabs 5)

The Venn diagram of the above datasets produced a list of 30 overlapped proteins as top candidates for integrated assessment via Rscore of relative abundance and tumor specificity in the above MS datasets as well as clinical association with OS at protein and mRNA expression levels via KMPlotter[35] and Protein Atlas[36]. We identified Plexin-B2 (encoded by *PLXNB2*, *PB2*) as the top-ranked oncoprotein in a negative association with patient OS across three clinical datasets (Fig. 1a, and Supplementary Fig. S1b-d, Supplementary Data 1-tabs 6-7). Of note, the list of the top 30 includes a previously characterized surface protein regulator of homotypic CTC cluster formation, CD44[16], within the most abundant proteins in cancer cells (Supplementary Fig. S1c), validating the potential relevance of listed adhesion molecules.

Our previous studies demonstrated that cancer cell-derived extracellular vesicles (EVs) and their surface proteins CD44 and CD81 contribute to the modulation of the microenvironment and EV-recipient cells[37]. To further assess the potential relevance of candidate proteins in cancer-specific secretome for cancer progression[38–41], we performed MS proteomic analyses of small EVs secreted by breast cancer cells (MDA-MB-231, SKBR3, BT4T4, and MCF7) and normal immortalized epithelial cells (HEK293, MCF10A and MCF12A). While most of the EV proteins were glycoproteins and membrane-bound proteins, PLXNB2 was also among the top enriched proteins in cancer EVs versus normal cell EVs (Supplementary Fig. S1e, and Supplementary Data 1-tabs 8–9). However, subsequent immunoblotting analysis revealed a truncated form of PLXNB2 (also known as β subunit[42]) in breast cancer cell-derived EVs which did not contain the full-length protein (Supplementary Fig. S1e), therefore the PLXNB2 functions in EVs were not further investigated and we focused on characterizing its cellular functions instead in this study.

### PLXNB2 expression is associated with unfavorable survival and enriched in CTC clusters

In KMPlotter[35] analysis of human breast cancer datasets[36], we found that high PLXNB2 expression is negatively associated with both OS and DMFS in all breast cancers and ER-negative breast cancers or grade 3 breast cancers (Fig. 1b–d, and Supplementary Fig. S1f and S1h). The other candidates following PLXNB2 include PPIA, EZR, and TLN1 (Supplementary Fig. S1g). We also analyzed a tissue microarray of advanced breast cancers collected at Northwestern University which showed a high expression of PLXNB2 across different breast cancer subtypes in association with distant metastasis (Supplementary Fig. S2a–b, and Supplementary Data 2).

We then measured the PLXNB2 expression in human blood CTCs and white blood cells (WBCs) from patients with stage III-IV breast cancer using multiple complementary methods[16,26]: FDA-approved

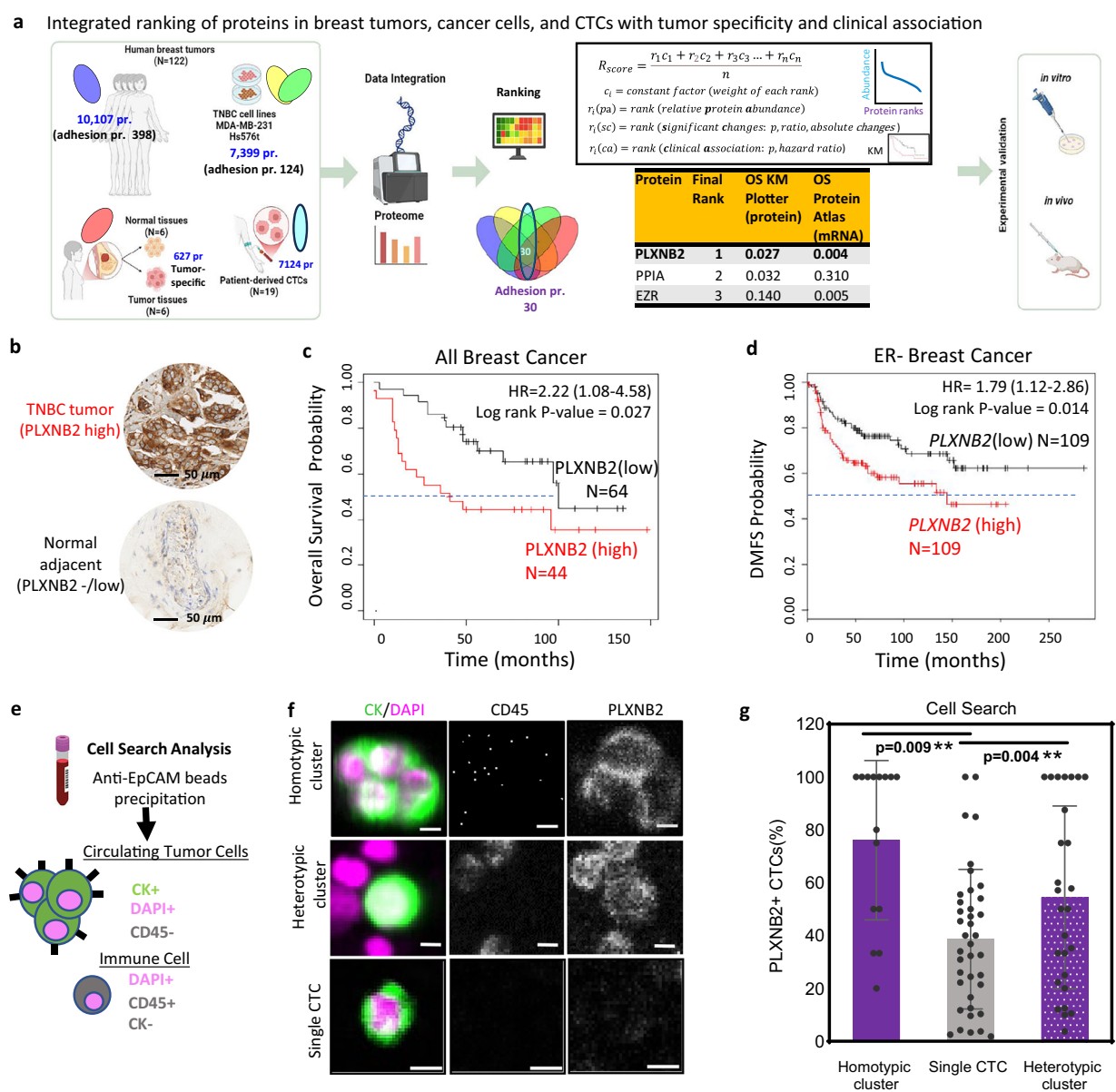

**a** Integrated ranking of proteins in breast tumors, cancer cells, and CTCs with tumor specificity and clinical association

**Fig. 1 | PLXNB2 expression in breast tumors is associated with poor prognosis and enriched in CTC clusters. a** A schematic of an integrated ranking, Rscore, of proteins in breast tumors, cancer cells, and CTCs with tumor specificity and clinical association, using multiple MS proteomic databases The mathematical model of Rscore integrates individual ranks ($r_i$) of each protein in (1) relative protein abundance, $r_i$(pc), in multiple datasets (patient tumors, CTCs, and cell lines), (2) significance changes, $r_i$(sc) in tumor specificity, including p-value, ratio or fold change, and absolute change, comparing TNBC voxels (laser capture microdissection) to normal adjacent tissues, and (3) clinical association, $r_i$(ca) including p value and hazard ratio, with OS and DMFS among multiple datasets. The significance of $r_i$ is multiplied by its constant weight factor ($c_i$) with the sum divided by n for an integrated Rscore and final top three candidates. Created in BioRender. Tong, F. (2025) https://BioRender.com/nea43wo. **b** Representative IHC images of PLXNB2high TNBC tumor and PLXNB2low normal breast tissue (adjacent to tumors) from a TNBC patient. **c** KM plot for OS of patients with all breast cancer in the

Tang_2018 data set (n = 108) via Kaplan-Meier plotter, separated by the best cut-off value of PLXNB2 protein expression (4) in primary tumors to define high vs. low within the expression range (0-11). *P* values were calculated via the Cox-Mantel (log-rank) test. **d** KM plot for DMFS of patients with ER⁻ breast cancer, divided by median cut-off of *PLXNB2* mRNA expression using data from GEO, EGA, and TCGA, n = 218. *P* values were calculated using a log-rank test. **e** Schematic depicting the patient blood sample workflow for CTC analysis on CellSearch. **f** Representative CellSearch images of a homotypic PLXNB2⁺ CTC-CTC (CD45⁻CK⁺DAPI⁺) cluster, a heterotypic PLXNB2⁺ CTC-WBC (CD45⁺CK⁻DAPI⁺) cluster, and a single PLXNB2⁻ CTC. Scale bar = 5 μm. **g** Portion (%) of PLXNB2⁺ CTCs in single CTCs in comparison with homotypic CTC clusters and heterotypic CTC-WBC clusters), respectively, analyzed via CellSearch as shown in (**f**), n = 41 patients. Data are presented as mean values +/- SD, *P* values reported are from two-sided unpaired t-tests unless specified. Source data are provided as a Source Data file.

CellSearch® with blood collected into fixative-containing CellSave tubes (N = 15 patients) (Fig. 1e, and Supplementary Fig. S2b–i), live cell flow cytometry[18,26] with blood drawn into EDTA tubes (N = 17 patients) (Supplementary Fig. S3a–b), immunoblotting (Supplementary Fig. S2d), and MS proteomic analysis (Supplementary Data 1-tabs 5). WBCs were recognized with the leukocyte marker CD45 expression.

Based on the CellSearch analyses, PLXNB2high breast cancer correlated with detectable blood CTCs, stained as cytokeratin (CK)⁺DAPI⁺CD45⁻ cells after enrichment via anti-EpCAM magnetic beads (Supplementary Fig. S2b–c). Compared to single CTCs, PLXNB2 expression was significantly higher in CTC clusters, both homotypic CTC-CTCs and heterotypic CTC-WBC clusters (Fig. 1f–g). Flow

cytometry analysis confirmed that a larger proportion of homotypic CTC clusters were PLXNB2-positive than single CTCs (Supplementary Fig. S3b–e). In orthotopic TNBC patient-derived xenografts (PDXs) labeled with luciferase 2-eGFP (L2G)[13], we observed a dynamic increase of surface PLXNB2 expression in CTCs and lung metastases in comparison to the primary tumor cells (Supplementary Fig. S3f). Furthermore, the detection of PLXNB2 on cell surface was resistant to trypsin digestion (Supplementary Fig. S3g). Breast tumor cells presented PLXNB2 in full-length (~200 kD) and truncated (~75 kD) proteins, both of which were lost upon CRISPR/Cas9 mediated gene knockout (KO) (Supplementary Fig. S2d).

## Loss of PLXNB2 reduces metastasis and CTC cluster formation in human breast tumors in vivo

PLXNB2 is a single-pass transmembrane plexin family member, and its primary function is to direct neural cell growth and migration in brain development; it also plays a role in the function of the vascular and endocrine systems, wound healing, monocyte function, and neuro progenitor cells[43–50]. However, PLXNB2's molecular mechanism was not previously studied in the context of CTC clusters and metastatic breast cancer. The existence of CTC clusters in patient blood has previously been shown to be associated with metastasis and reduced OS in breast cancer[11,22]. Since our work revealed that PLXNB2 is highly expressed in patient CTCs, we continued to determine if PLXNB2 promotes CTC cluster formation and spontaneous metastasis in vivo.

Taking advantage of the CRISPR/Cas9 technologies and the target protein detection via flow cytometry, we transduced the MDA-MB-231 cells with lentiviral *PLXNB2* gRNAs and then sorted multiple pools of *PLXNB2* knockout cells (KO1 pool 1 and KO2 pool 2) without clonal selection based on the negative expression of the surface receptor PLXNB2 within 72 hours. These *PLXNB2* KO cells did not significantly alter cellular growth (confluence), viability, cell migration, and invasion in vitro as well as 10,000 tumor cell-mediated growth curves in vivo (Supplementary Fig. S4a–h) whereas the tumorigenesis in serial dilutions of 10 to 1000 cell implants was compromised (Supplementary Fig. s4i–k). Notably, the tumor growth phenotype of stable *PLXNB2* KO tumor cells (via Cas9/gRNAs) might have resulted from selective pressure in KO cell maintenance, differing from the variable effects of siRNA-mediated transient KD of *PLXNB2* in our following studies (shown in Supplementary Fig. S5) as well as the reported inhibitory effects of shRNA-mediated *PLXNB2* knockdown (KD) on tumor cell proliferation[51]. We speculate that the CRISPR Cas9-generated KO and selective growth pressure resulted in a compensation of pathways for the restored proliferation of these cells.

After orthotopically implanting 10,000 tumor cells into the fourth mammary fat pads of NSG mice to ensure tumor growth of both L2G-labeled WT (pool control clonalities) and *PLXNB2* KO MDA-MB-231 cells (pooled KO cells), we assessed spontaneous metastasis of these tumors to the lungs (Fig. 2a). Mice were monitored for 8–10 weeks until the experimental endpoint for collections of tumors, lungs, and blood. Compared to control tumors, the *PLXNB2* KO tumors did not show a significant difference in Ki67-indicated proliferation ($P = 0.14$) but a trend of lighter tumor weight ($P = 0.06$) (Fig. 2b, c, and Supplementary Fig. S4c–e). These data suggest that stromal factors and host interactions with KO tumor cells might contribute to the borderline changes in tumor burden in vivo. The mouse lungs, however, showed a significant 23-fold reduction of the metastatic burden from *PLXNB2* KO tumors compared to those of control tumors (Fig. 2d–e), which were normalized by tumor weight. L2G$^+$ CTCs were analyzed after blood collection via heart puncture from each mouse and by H&E staining analyses of vascular CTC in situ within the lung sections. *PB2* KO reduced the CTC cluster formation, as detected from the blood and within lung tissue sections; and *PB2* KO inhibited spontaneous metastatic colonization (metastatic lesions and cell numbers) (Fig. 2f–h, and Supplementary Fig. S4f). These data demonstrated that *PLXNB2*

depletion inhibits CTC cluster formation and blocks spontaneous metastasis of human breast cancer in vivo.

To determine if PLXNB2-mediated CTC clustering was associated with increased co-colonization in metastasis, we orthotopically implanted red L2T- and green L2G-labeled *PLXNB2*$^+$ WT control tumors (ConT and ConG) and *PB2*$^-$ KO tumors (KOT and KOG) into separate left and right 4$^{th}$ mammary glands with 4 groups of combinations: (1) ConT-ConG, (2) KOT-KOG, (3) ConT-KOG, and (4) KOT-ConG (Fig. 2i). After 6 weeks of orthotopic tumor growth, only the mice bearing the ConT-ConG tumors in dual colors showed dual-color lung colonies whereas the counts of both single-color and dual-color metastatic colonies dramatically decreased in any of the three groups with one or two KO tumor implants (Fig. 2j-m). The CTC clusters were dramatically in higher frequencies in control tumor-bearing mice than the KO tumor-bearing mice (Fig. 2m). These data are in consistency with previous demonstrations that lung co-colonization (polyclonality) are contributed by CTC clusters of breast cancer[16] and pancreatic cancer[52], albeit a possibility of sequential seeding of polyclonal tumor cells.

To explore the effects of *PB2* KO on animal survival, we found that the control tumor-bearing mice survived up to 8 weeks whereas the KO tumor-bearing mice went up to 10 weeks with relatively comparable tumors before they exceeded the allowable tumor sizes for euthanasia (Fig. 2n). Nevertheless, the blood CTC clusters and lung metastases from the control tumors remained over 30 times higher than those from the KO tumors in mice (Fig. 2o–p). Notably, the majority of the CTCs from both control and KO tumor groups were arrested in G0/G1 phases without significant proliferation differences (Fig. 2o), in consistency with comparable Ki67-indicated proliferation of lung metastases of these tumors (Supplementary Fig. S4e). These results suggest that *PLXNB2* depletion-reduced metastases are independent of its proliferation effects on the CTCs, or the cells disseminated into the lungs.

## PLXNB2 depletion compromises tumor cell clustering and mammosphere formation

After identifying PLXNB2's role in promoting CTC clusters and metastasis, we continued to determine its function in tumor cell clustering in specific breast cancer subtypes such as TNBC and HER2$^+$ in which metastasis is common. To do this, we sorted L2G-labeled PLXNB2$^{high}$ and PLXNB2$^{low/-}$ tumor cells from orthotopic TNBC PDX models (TN3) which develop spontaneous lung micro-metastases[13] (Fig. 3a). Next, using the IncuCyte Live Cell Imager® as previously described[16], the clustering of primary PDX tumor cells on collagen-coated plates was monitored over time. After 6–8 h of clustering, PLXNB2$^{high}$ cells formed significantly more clusters (> 2-3 cells) than PLXNB2$^{low}$ cells (Fig. 3b, c). We then transiently knocked down *PLXNB2* in multiple TNBC cell lines (human MDA-MB-231 and HS578T, and mouse 4T1) as well as HER2$^+$ cell line SKBR3, using both Smart-Pool siRNA (si*PB2*) and individual siRNAs (si*PB2*−09, −10, −11). The reduction or loss of the full-length and truncated PLXNB2 was validated by immunoblotting and/or flow cytometry (Supplementary Fig. S5a, S5c–d, S5j). MDA-MB-231 cells transfected with si*PB2* started to show slower growth (confluence) than the control cells at 24-48 h after seeding, whereas si*PB2*−10 did not have a significant effect on the growth (Supplementary Fig. S5a). However, over a short period of 4-6 h without compromised cell viability and with minimal influence by cell proliferation/growth, both si*PB2* and si*PB2*−10 comparably reduced the size of tumor clusters of all five tested models, including human MDA-MB-231, MDA-MB-468, SKBR3, and HS578T cells, and mouse 4T1 cells (Fig. 3d–g, and Supplementary Fig. S5a-n), suggesting that PLXNB2 is required for tumor cell clustering in breast cancers.

Consistent with the effects of si*PB2* and si*PB2*−10 (Supplementary Fig. S5), both *PB2*-KO1 and -KO2 cell pools depleted tumor cell clustering (Supplementary Fig. S6a–b). However, *PB2* KO cells did not

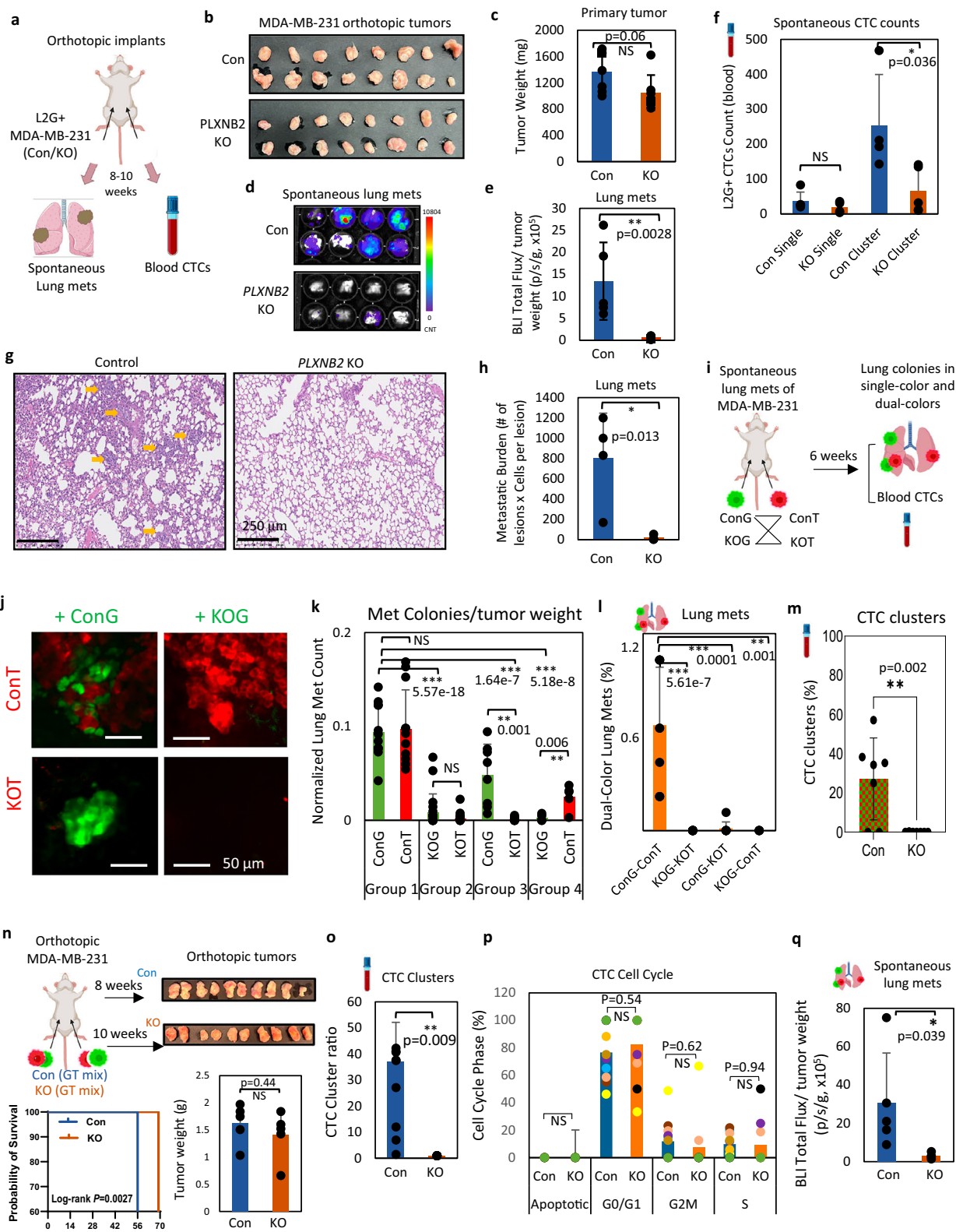

significantly alter the FAK phosphorylation, cell growth (confluence), and cell motility (migration and invasion) in vitro (Supplementary Fig. S6c–e), possibly due to selective pressure. These results demonstrate that PLXNB2 is necessary for tumor cell clustering but acts independently of the phenotypes on proliferation or cell motility.

PLXNB2 is a large single-pass transmembrane protein made up of multiple domains[53] (Fig. 3h): the extracellular region (ECTO) including

a semaphorin domain (Sema) responsible for binding to receptor/ligands, three Ig-like fold domains and three plexin-semaphorin-integrin domains; a transmembrane domain; the intracellular Rho-Binding Domain (RBD); the intracellular GTPase activating protein domain, and the intracellular PDZ-binding domain (VTDL). Using the *PB2* mutant constructs shared by the Friedel Lab[54], we overexpressed the full-length *PLXNB2* and its various mutant forms (dECTO, mRBD,

**Fig. 2 | PLXNB2 depletion abolishes spontaneous metastasis and CTC cluster formation in TNBC in vivo. a** Schematic showing the experimental workflow of orthotopic implantation and analyses of lung metastases and CTCs.
**b, c** Representative images (**b**) and weight quantification of *PLXNB2* Con and KO tumors (**c**) from mice at 10 weeks, n = 8 mice/group. **d, e** Bioluminescence images (BLI) of mouse lungs ex vivo (**d**) and quantified lung metastasis (**e**) at 10 weeks, n = 8 mice/group. **f** L2G+ CTC counts (single and clusters) detected in *PLXNB2* Con and KO mouse blood at 10 weeks of spontaneous metastasis, n = 4 mice/group.
**g, h** Representative images (**g**) and metastatic burden (**h**) in *PLXNB2* Con and KO metastatic cells of mouse lungs with H&E staining, scale bar = 250 μm; experiments were repeated with the PB2 KO clone, n = 4 lungs/group. Yellow arrows point to the micro metastases in the lungs. **i** Schematic representing experimental workflow of dual color implantations of L2T+ (red) or L2G+ (green) MDA-MB-231 tumor cells with *PLXNB2* control (ConT, ConG) or *PLXNB2* KO (KOT, KOG); mice were sacrificed after 6 weeks for analyses, n = 4 mice/group. **j** Representative images of green (ConG/

KOG) and red (ConT/KOT) MDA-MB-231 colonies in the lungs of mice, n = 4 mice/group. **k** Counts of L2T+ or L2G+ colonies in the lungs of mice after 6 weeks orthopedic, n = 4 mice/group. **l** Counts of dual color colonies with red and green tumor cells in the lungs, n = 4 lungs/group. **m** CTC clusters in red/green count as analyzed via flow cytometry in the mice bearing the *PLXNB2* Control and KO tumors, n = 8 mice/group. **n** Top left: Schematic of orthotopic implantation of *PLXNB2* Con and KO tumor cells into NSG mice (200,000 cells/site). Bottom left: KM plot of the mice shows Supplementary survival (2 weeks) in the mice bearing Con vs. KO tumors. Right panels: Photos (top) and bar graphs (bottom) of relatively comparable tumor weight between Con (8-week) and KO groups (10-week), n = 5 mice/group. **o–q** Bar graphs of blood CTC clusters (**o**), cell cycle phases (**p**), and spontaneous lung metastasis (**q**) between the Con and KO tumors, n = 5 mice/group. Data are presented as mean values +/-SD, with P values reported from two-sided unpaired t-testsunless specified. Source data are provided as a Source Data file.

and dVTDL) back into *PB2* KO cells and validated the expression via immunoblotting (Fig. 3i) or flow cytometry (Supplementary Fig. S6f). While full-length *PLXNB2* overexpression effectively rescued the clustering of *PB2* KO cells, the *PB2* mutant depleting the ECTO domain (dECTO) had the lowest cluster-rescuing function, followed by the mutants depleting or modifying the intracellular signaling domains (dVTDL and mRBD) (Fig. 3j, k). This data demonstrates that both extracellular and intracellular signaling properties in full-length *PLXNB2* are required to promote tumor cell clustering.

As self-renewal is often required for CTCs to regenerate tumors[13,55–57],we next determined if PLXNB2 regulates self renewal-related tumorigenesis in vivo and mammosphere formation of breast cancer cells in vitro[16]. We found that of *PB2* depletion via KO or KD significantly compromised the orthotopic tumorigenesis at serial dilutions (down to 10 cells) with reduced frequencies of tumor-initiating cells (TICs) and reduced the number of mammospheres formed in TNBC cells (Supplementary Fig. S4i, j, S6g, h). The mammosphere phenotype in *PB2* KO cells was restored by overexpression of the full-length *PLXNB2* and to a lesser extent by overexpressed dECTO *PB2* mutant, suggesting that the extracellular domain of PLXNB2 is necessary for its functions in promoting self-renewal of tumor cells (Supplementary Fig. S6i-j).

Given that tumor cell clusters are powerful precursors to metastatic colonization, we wanted to determine if PLXNB2 levels would impact tumor cell seeding to the lungs. Within 4 days after tail vein injection for experimental colonization, the mice injected with L2G-labeled[13] *PB2* KD cells (MDA-MB-231) either by si*PB2* or si-*PB2*–11 had shown 80-90% less disseminated colonization to the lungs compared to the control group (Fig. 3l-o). These data suggest that PLXNB2 promotes tumor clustering, self-renewal, and experimental lung colonization, contributing to metastasis.

## PLXNB2 signals through SEMA4C to promote tumor cell clustering

To elucidate the molecular mechanism by which PLXNB2 facilitates tumor cluster formation, we investigated whether in breast cancer PLXNB2 interacts with any of its canonical ligands on the cell surface, such as single-pass transmembrane proteins, semaphorin (Sema) family members SEMA4A, 4C, 4D, and 4G[51,54,58–62]. From our ranked list of adhesion proteins, we found expression of SEMA4C, SEMA4D, and SEMA4A in treatment-naïve breast cancer[32] (Supplementary Fig. S6k); however, only SEMA4C was detected in breast cancer cells (human/mouse cell lines and/or PDX tumor cells) via immunoblot analyses and mass spectrometry (Fig. 4a, b, and Supplementary Data 1). Based on publicly available databases, we also found that high *SEMA4C* mRNA expression in breast cancer correlates with unfavorable DMFS and OS (Supplementary Fig. S6l-m). We hypothesized that SEMA4C is a primary ligand of PLXNB2 in driving homotypic tumor cell clusters in metastasis.

To confirm if Sema molecules interact with PLXNB2 in breast cancer cells, we performed a co-immunoprecipitation for human PLXNB2 using clustered TNBC cells; and found that SEMA4C was specifically enriched in the PLXNB2 protein complex, whereas SEMA4D was nearly undetectable (Fig. 4c), demonstrating SEMA4C-PLXNB2 interactions in the homotypic CTC clusters. In the anti-PLXNB2 pull-down protein complex, we also detected CDC42 (Fig. 4c), which is a Rho GTPase family member known to be regulated by PLXNB2 signaling[63,64].

To determine the functional importance of SEMA4C in tumor clustering, we knocked down *SEMA4C* in the breast cancer cells (Fig. 4d). *SEMA4C* KD alone mimicked *PB2* KO in reducing the average size and numbers of WT tumor cell clusters, whereas loss of both *PLXNB2* and *SEMA4C* together did not have an additional impact on cluster size compared to *PB2* KO alone (Fig. 4e–g). This indicates that these two interacting proteins might belong to the same interaction pathway necessary for homotypic tumor clustering.

To gain a better understanding of the downstream molecules and pathways in PLXNB2-mediated tumor cell clustering, we performed an MS-based global proteomics analysis of MDA-MB-231 breast cancer cells in suspension before and after clustering. Principal component analysis indicates three groups of samples and Heatmap comparisons show 6 distinct groups of protein signatures, up-regulated or down-regulated within 4 h of clustering or by si*PB2*-mediated KD (Fig. 4h-i, Supplementary Data 3). Additional GO analysis of the altered proteins in si*PB2* cells includes biological processes in chromosome condensation, cell cycle, RNA export, DNA methylation, and others (Fig. 4j). Groups C and F, with clustering-specific up-regulation and down-regulation in protein expression, respectively, identified pathways in protein folding and hydroxylation, UV protection, and response to gamma radiation, all of which were over 10-fold enriched in PLXNB2+ tumor cell clusters (Supplementary Fig. S7a and Supplementary Table S1).

In search of PLXNB2 downstream target proteins related to TNBC phenotypes, we found that MCM7, an essential component of the DNA helicase, was one of the top candidates significantly decreased in the cells with si*PB2*-mediated KD (Supplementary Fig. S7b). MCM7 has previously been implicated in cancer stemness and metastasis[65], thus making it an interesting target for clustering studies. TNBC cells transfected with siMCM7 for gene KD showed a compromised clustering within 6 h (Supplementary Fig. S7c–d). Thus, our data demonstrates a mechanism by which PLXNB2 can promote clustering through influencing MCM7. However, MCM7 overexpression did not rescue the clustering nor the mammosphere formation of TNBC cells with *PB2* KD or KO (Supplementary Fig. S7e–h), indicating that MCM7 is necessary but not sufficient to rescue the PLXNB2 signaling in clustering and self-renewal. Moreover, KD of CDC42, a known target of PLXNB2 signaling[64] and a component of the PLXNB2 protein complex (Fig. 4c), mimicked the KD of PLXNB2 and MCM7 in reducing

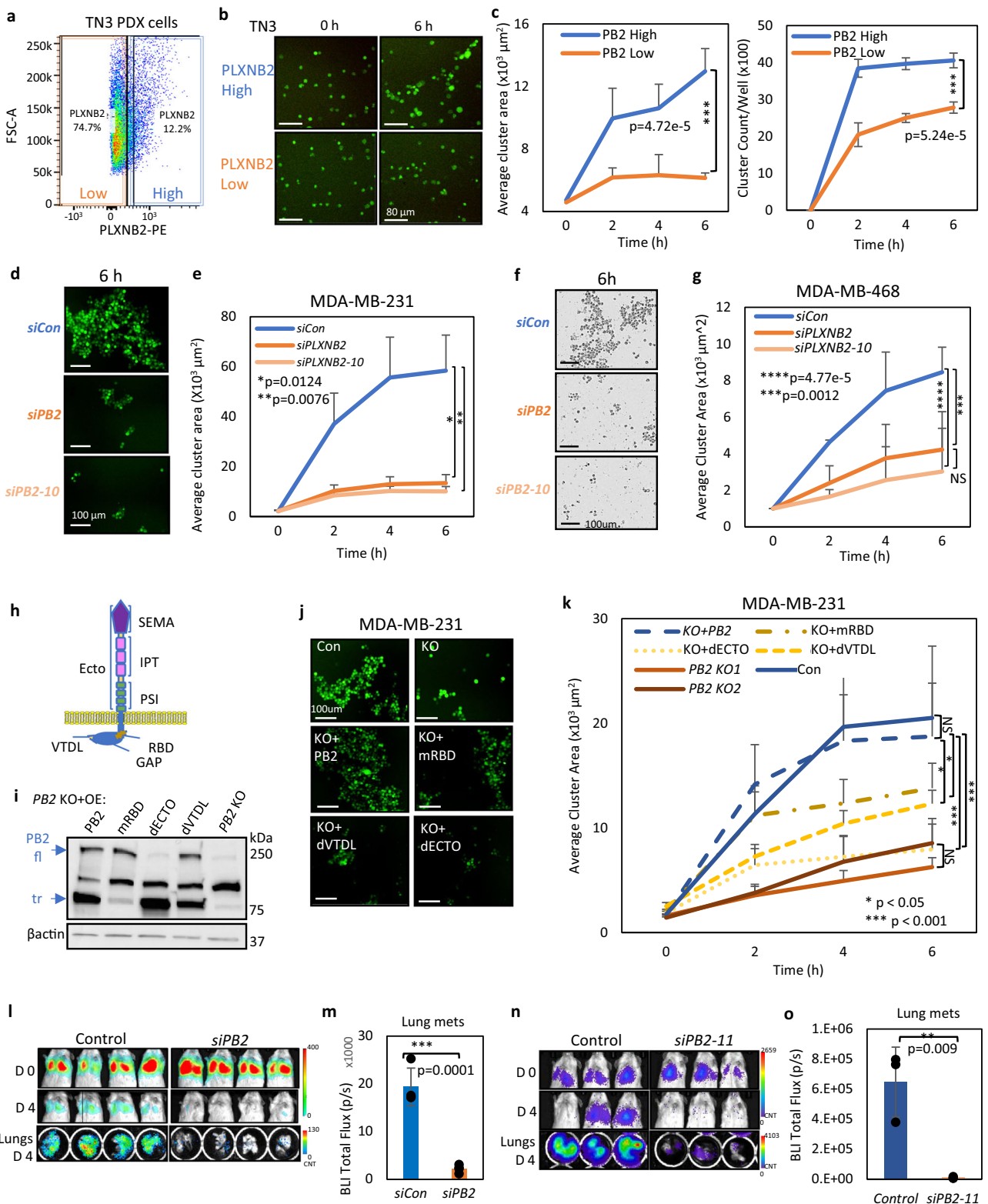

efficiencies of mammosphere formation and tumor cell clustering in vitro (Supplementary Fig. S7i–k).

## PLXNB2-dependent heterotypic clustering signals through SEMA4A on monocytes

In addition to MCM7, the global mass spectrometry analysis of *PB2* KD cells revealed many biological processes at 10- to 40-fold enrichment, such as positive regulation of the immune response to tumor cells and astrocyte activation (Supplementary Table S1). Based on our Cell-Search data that demonstrated PLXNB2 enrichment in the CTC-WBC heterotypic clusters (Fig. 1f-g), we hypothesized that PLXNB2 in tumor cells impacts interactions with immune cells. Since immune cells, especially myeloid cells, have been found in heterotypic CTC clusters driving cell cycle progression, proliferation, and/or immune evasion[19,66], we proposed to determine what, if any, role PLXNB2 plays in heterotypic CTC-immune cell clusters (Fig. 5a).

**Fig. 3 | PLXNB2 promotes tumor cell clustering and metastatic dissemination.**
**a** Flow panel showing the gating of PLXNB2⁺ and PLXNB2⁻ cells sorted from dissociated primary TN3 L2G⁺ PDX cells for clustering in (**b**, **c**). **b**, **c** Representative images of clustering (**b**) and average cluster area and cluster count curves of sorted PLXNB2 high and low TN3 PDX cells (**c**); $n = 5$ technical replicates examined over 3 independent experiments. P-value was calculated using two-sided unpaired t-test. **d**, **e** Representative images (**d**) and cluster area (**e**) of MDA-MB-231 cells transfected with siRNA control (siCon), *PLXNB2* SmartPool siRNA (si*PLXNB2*), and single siRNA (si*PLXNB2*-10), $n = 5$ technical replicates examined over 3 independent experiments. P-value was calculated using two-sided unpaired t-test. **f**, **g** Representative images (**f**) and cluster area (**g**) of MDA-MB-468 cells transfected with siCon, si*PLXNB2*, and si*PLXNB2*-10, $n = 5$ technical replicates examined over 3 independent experiments. *P* values were calculated using one-sided ANOVA. **h** Schematic of PLXNB2 domains: Extracellular (Ecto) domains include SEMA = SEMAphorin domain; IPT = Ig-like fold domain; and PSI = Plexin-SEMAphorin-integrin domain. Intracellular domains include RBD = Rho-binding domain; GAP = GTPase activating

protein domain; and VTDL = PDZ-domain binding site (Rho-GEF binding).
**i** Immunoblots of overexpressed *PB2* mutants with either full-length (fl) or truncated (tr) depletions: mutated RBD (mRBD) tr, depleted Ecto domain (dECTO) fl and depleted VTDL (dVTDL) tr in MDA-MB-231 *PB2* KO cells. **j**, **k** Representative images (**j**) and cluster area growth curves (**k**) of MDA-MB-231 *PB2* KO clusters with overexpression of *PLXNB2* full-length or mutants (mRBD, dECTO, or dVTDL); Con vs. KO + *PLXNB2* p = 0.67, Con vs. KO1 $p = 0.0002$, Con vs. KO2 $p = 1.43e-6$, KO + *PLXNB2* vs. KO+mRBD p = 0.04, KO + *PLXNB2* vs. KO+dECTO p = 1.69e-6, KO + PB2 vs. KO+dVTDL p = 0.005, KO + *PLXNB2* vs. KO1 p = 0.0002, $n = 5$ technical replicates examined over 3 independent experiments. *P* values were calculated using one-sided ANOVA. **l–o** Bioluminescence images (**l**, **n**) and quantified BLI signals (**m**, **o**) of dissected lungs ex vivo after transfections with siCon, si*PLXNB2* (**l**), or si*PLXNB2*-11 (**n**), $n = 4$ mice/group (**l**, **m**) and $n = 3$ mice/group (**n**, **o**). P-value between two groups was calculated using two-sided unpaired t-test. Data are presented as mean values ± SD. Source data are provided as a Source Data file.

In co-culture suspension, we mixed WT or *PB2*-KO MDA-MB-231 cells with patient WBCs. The *PB2* WT cells formed larger and more heterotypic clusters with WBCs than *PB2* KO cells without affecting tumor cell viability (Fig. 5b–c, and Supplementary Fig. S4b, S8a), demonstrating that PLXNB2 promotes heterotypic tumor-WBC cluster formation in which tumor cells can possibly evade immune cell killing. To identify which ligand might be involved in CTC-WBC cluster formation, we used a computational ranking of surface proteins to determine which canonical ligands of PLXNB2, i.e., semaphorin interactors[67], were expressed in WBCs. The online single-cell RNA sequencing data suggested that among canonical PLXNB2 ligands, SEMA4A had the highest expression within peripheral blood derived mononuclear cells (PBMCs) (monocytes), whereas SEMA4C was minimally expressed (https://www.proteinatlas.org) (Supplementary Fig. S8b). Consistently, the published MS proteomic data[67] also revealed a higher abundance of SEMA4A protein than SEMA4D with undetectable SEMA4C in human blood monocytes (Supplementary Fig. S8b, and Supplementary Data 1). Furthermore, our flow cytometry analyses of the blood cells from patients with breast cancer demonstrated that, compared to granulocytes and lymphocytes, monocytes express a higher level of SEMA4A (Fig. 5d, and Supplementary Fig. S8c), and that the heterotypic CTC-WBC clusters double positive for EpCAM and CD45 have enriched co-expression of PLXNB2 and SEMA4A versus WBC-only clusters and single cells (Fig. 5e, Supplementary Fig. S8d-e), implying the possible contribution of PLXNB2 and SEMA4A to heterotypic CTC-WBC clustering.

To determine the protein interactions between monocyte SEMA4A and tumor cell PLXNB2, we utilized the THP1 monocytic cells which express high SEMA4A ( > 80%) (Supplementary Fig. S8f) as an alternative source of human monocytes for heterotypic cluster formation. Similar to patient WBCs, they do not impact the viability of *PB2* KO tumor cells in co-culture (Supplementary Fig. S8g). Using human tumor cell-THP1 clusters at an optimized 1:4 ratio and anti-PLXNB2 antibody for co-immunoprecipitation, we also detected the enrichment of SEMA4A and CDC42 in the PLXNB2 protein complex, compared to monocytes only (Fig. 5f, g).

To determine the importance of PLXNB2 in tumor-monocyte clustering, we found that in mixed cell suspension, PLXNB2 control tumor cells formed clusters effectively with THP1 cells (1:4 ratio), whereas *PB2* KO TNBC cells lost the capability for heterotypic cluster formation (Fig. 4h-i). Consistently, sorted primary PDX TNBC cells (TN1) with PLXNB2^high expression formed significantly larger heterotypic clusters with THP1 cells than PLXNB2^low TN1 cells, when co-cultured under the previously established clustering conditions on collagen-coated plates[16] (Fig. 4j-k, Supplementary Fig. S8h).

To further determine the association of SEMA4A with heterotypic tumor-monocyte clustering, we sorted SEMA4A⁺ and SEMA4A⁻

THP1 cells and genetically knocked down *SEMA4A* in THP1 cells for co-culture with *PB2* WT and KO TNBC cells. SEMA4A⁺ monocytes with PLXNB2⁺ tumor cells (double positive) formed the largest clusters with more cell counts per cluster and in highest numbers of clusters compared to single-negative or double-negative combinations of SEMA4A⁻/⁺ monocytes with PLXNB2⁺/⁻ tumor cells in heterotypic clustering (Fig. 5l, m, Supplementary Fig. S8i, and S9a–b). Interestingly, loss of *SEMA4A* alone in monocytes was sufficient to reduce the size of heterotypic PLXNB2⁺ tumor clusters which is comparable with the effects of *PB2* KO in tumor cells, suggesting that SEMA4A is the primary ligand on monocytes for PLXNB2-dependent tumor-monocyte clustering (Fig. 5a).

We then compared the outcomes of experimental colonization with TNBC (MDA-MB-231) control cells and *PB2* KO cells which were pre-clustered for 4 h ex vivo prior to tail vein-injections. With minimal clustering capacity, the *PB2* KO cells showed a significant reduction in tumor cell dissemination to the lungs after 12 h, and the phenotype was maintained for up to four days (Fig. 5n), suggesting that PLXNB2 enhances dissemination and metastatic colonization, coupled with tumor clustering and independent of proliferation effects. Similarly, after 4 h pre-clustering with unlabeled THP1 monocytes, heterotypic L2G⁺ tumor cell-THP1 clusters promoted tumor cell dissemination (12 h), in a *PLXNB2*-dependent manner (Fig. 5n, o). Meanwhile, the presence of THP1 monocytes promoted the metastatic colonization of both WT and *PB2* KO tumor cells within 4 days (Fig. 5n, o), possibly through both PLXNB2-dependent clustering and clustering-independent factors, such as stemness[68]. These results demonstrated that monocytes promote heterotypic CTC clustering, early dissemination, and lung colonization.

**Loss of Plxnb2 from mouse tumor cells reduces metastasis and CTC-WBC formation in vivo**
Finally, to determine the role of mouse *Plxnb2* in CTC-immune cell cluster formation, we knocked down *Plxnb2* gene expression in mouse 4T1 TNBC cells with si*Plxnb2* transfections and found that loss of Plexin-B2 in tumor cells diminished the heterotypic 4T1-mouse WBC cluster formation (Fig. 6a–b). 6 h after the L2T⁺ tumor cells were inoculated into the tail vein of immune competent Balb/c mice, the blood CTC analyses also revealed that *Plxnb2* KD reduced heterotypic CTC clustering with monocytes (Fig. 6c), which possibly provided opportunities for other WBCs (such as T cells) to interact with 4T1 tumor cells (Supplementary Fig. S9c-e).

To assess the effects of Plexin-B2 depletion on spontaneous metastasis in immune competent mice, we further generated *Plxnb2*-KO 4T1 cells using CRISPR/Cas9 technologies and orthotopically implanted the WT control cells (Control) and KO cells (1.5 × 10⁶ cells) into the 4th mammary fat pads of Balb/c mice (Fig. 6d). By Day 9 when

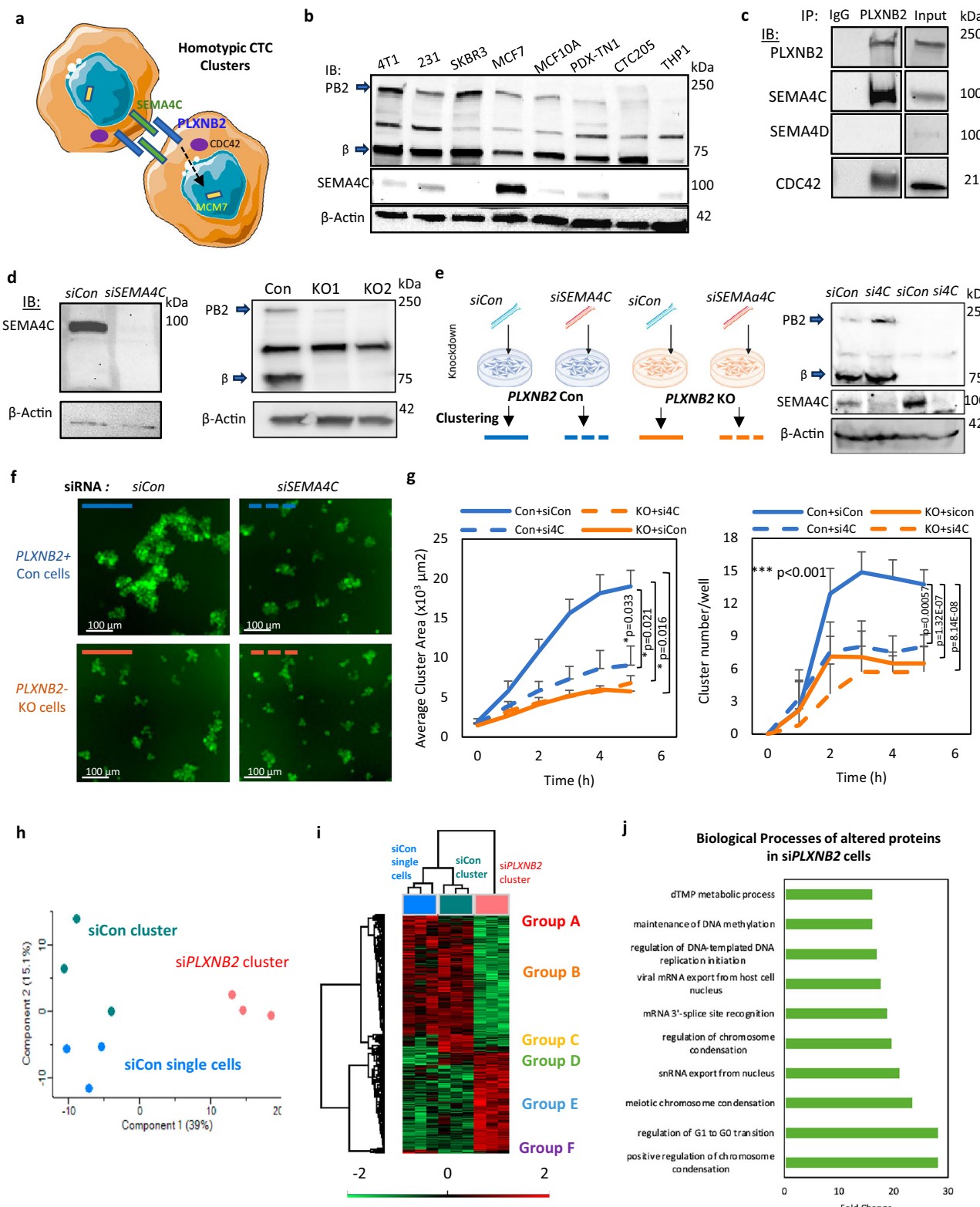

two groups of tumors developed into comparable sizes (tumor weight), the *Plxnb2*-KO tumor group significantly reduced spontaneous metastases to the lungs, coupled with decreased CTCs, both singles and CTC-WBC clusters, particularly the CTC-myeloid clusters (Fig. 6e–i), as analyzed by bioluminescence imaging and flow cytometry. To further confirm the lung metastasis measured via IVIS bioluminescence/fluorescence imaging (Fig. 6e), we performed histology

analyses of the mouse lungs, which cross-validated that *Plxnb2* depletion via CRISPR KO in 4T1 cells reduces the spontaneous micrometastic lesions of 4T1 tumors spread to the mouse lungs (Supplementary Fig S9f–g).

Taken together, our data demonstrates that mouse *Plxnb2* and human *PLXNB2* play a similar role in promoting CTC-monocyte cluster formation and driving spontaneous metastases in breast cancer.

**Fig. 4 | PLXNB2 interacts with SEMA4C to promote homotypic clustering of breast tumor cells. a** Schematic of PLXNB2-SEMA4C intercellular interactions in homotypic tumor cell clustering between two cancer cells. **b** Immunoblots of PLXNB2 (PB2) and SEMA4C expression in different breast cancer cell lines and PDX models using antibodies that are specific to detect both human and mouse iso-forms. **c** Immunoblots of SEMA4C, CDC42, and PLXNB2 in the protein complex immunoprecipitated by anti-PLXNB2 antibody compared to IgG. **d** Immunoblot images showing reduced SEMA4C levels after *SEMA4C* KD in MDA-MB-231 tumor cells (left) and depleted *PLXNB2* in *PLXNB2* KO1 and KO2 cells (right). **e** Schematic illustration and immunoblotting of MDA-MB-231 control and *PLXNB2*-KO cells transfected with siRNA control or *siSEMA4C* knockdown, followed by tumor cell clustering assessment in suspension. **f, g** Representative images (**f**) and average cluster area/size curves (**g**) of MDA-MB-231 control (Con) and *PB2* KO (KO) cells transfected with control siRNA (siCon) or *SEMA4C* siRNAs (si4C) for KD, measured by IncuCyte, *n* = 5 technical replicates examined over 5 independent experiments.

For cluster size (area) comparisons with the group of Con+siCon, Student's t test *p* = 0.033 for Con+si4C, *p* = 0.021 for KO+si4C, and *p* = 0.016 for KO+siCon. For the cluster number comparisons with the Control group, *p* < 0.001 for all three pairs above. *P* value were calculated using one-sided ANOVA. **h** Principal component analysis clusters of MDA-MB-231 single cells, siCon clusters, and si*PLXNB2* clusters analyzed from global proteomics analysis with *n* = 3 technical replicates/ group over three independent experiments. **i** Heat map of differentially expressed pro-teins from global proteomics analysis of siCon single cells (blue), siCon clusters (green), and si*PLXNB2* clusters (red), with *n* = 3 technical replicates/ group over three independent experiments. **j** Gene ontology biological processes analysis of significantly up-regulated and down-regulated proteins in *PLXNB2* control vs. *PLXNB2* KD clusters in breast cancer cells analyzed by mass spectrometry. Data are presented as mean values ± SD. *P* values were calculated using one-sided ANOVA. Source data are provided as a Source Data file.

## Discussion

Our studies have established an unprecedented computational rank-ing of quantitative global MS-based relative protein abundance and tumor-specific expression and integrated it with clinical outcome analyses and experimental function determination. PLXNB2 is identi-fied as a driver that promotes both homotypic and heterotypic CTC clustering through its interactions with Semaphorin ligands SEMA4C on tumor cells and SEMA4A on monocytes, in addition to PLXNB2 functions in proliferation and other known functions[43–50]. It has been reported by Gurrapu et al. that PLXNB2 promotes breast cancer cell proliferation through the RhoA and MAPK signaling pathways[51], which appears to be context-dependent and influenced by the expression levels of co-receptors such as MET and ErbB2. Furthermore, our stu-dies reveal proteins such as MCM7 as downstream mediators of the PLXNB2 interaction network that are required for PLXNB2-mediated tumor cluster formation but might not be sufficient to rescue surface protein-mediated intercellular crosstalk. Future therapeutics aimed at blocking PLXNB2 and SEMA4C/4A interactions can potentially serve as targeting strategies in breast cancer, especially TNBC and complement existing treatments, such as pembrolizumab[69–71], Sacituzumab[72,73], and Palbociclib[74] for improved outcomes.

CTC clusters may contain multiple cell types. Homotypic tumor cell clusters are often enriched with tumor cells that have stem cell properties[16,26,37,75], whereas heterotypic clusters contain mixtures of tumor and immune cells in which the tumor cells may evade immune cell attack and gain proliferation advantages[19,76]. Our previous work and that of others have identified a few molecular drivers of homotypic tumor cell clusters, including tumor-initiating cell markers and receptors[16,26,37] and molecules in cell junctions or adhesive structures[11]. The observation that heterotypic CTC-monocyte clusters enhance metastatic colonization highlights a potential pro-metastatic role for monocytes in facilitating tumor cell dissemination and colonization. Similarly, neutrophils within heterotypic CTC clusters promote cell cycle progression and metastatic potential of CTCs in breast cancer[19].

CTC clusters have up to 50-fold greater metastatic potential compared to single CTCs, making it a very crucial population to target. While it has long been hypothesized that CTCs can also cluster with immune cells, limited research has thus far been devoted to identifying the molecular markers of these heterotypic clusters. In addition to myeloid cells such as the monocytes specified in this work, neutrophils, MDSCs[19,77], and the plethora of other immune cell types need to be further investigated for potential interactions with and influences on CTCs in cancer metastasis.

We demonstrate a strong correlation between PLXNB2-mediated CTC cluster formation (homotypic and heterotypic) and lung metas-tasis. While these findings suggest that PLXNB2 plays a crucial role in promoting metastasis through CTC clustering, we acknowledge that metastasis is a highly complex and multifaceted process. Prior studies have shown that polyclonal metastases can also arise through

sequential seeding events[52]. Metastasis is a series of complex steps by which tumor cells shed off the primary tumor, enter the vasculature (intravasate), circulate, and ultimately extravasate into secondary tissues[14,78,79]. Previous studies demonstrate that myeloid cells such as macrophages promote migrating tumor cell contact with the vascu-lature and enhance vascular permeability for intravasation[80,81]. Con-sidering the tumor-myeloid interactions promoted by PLXNB2, PLXNB2 may also promote intravasation of tumor cells in addition to CTC dissemination.

From the mass spectrometry analyses, the unexpected identifi-cation of the tumor cell PLXNB2 signaling pathway in connection with astrocyte activation and immune response might imply its role in regulating the immune microenvironment of central nervous system (CNS) and other organs. PLXNB2 is known to regulate neuronal migration and synaptogenesis[46,82] and has previously been implicated in regulating cellular biomechanics, stemness, host microenviron-ment, and angiogenesis in brain cancer and other cancers[43,51,54,62,83,84]. The newly identified role of PLXNB2 in CTC clustering in conjunction with its ligands SEMA4C in tumor cells and SEMA4A in monocytes promotes metastasis, and is correlated with lower OS and DMFS in the context of advanced breast cancer, TNBC in particular. It is likely that certain cancer types or host immunity[85,86] may confound the effects of PLXNB2[87]. The context-dependent functions and biomarker associa-tion of PLXNB2 may be largely dependent on the ligands it interacts with and the microenvironmental conditions[36,88]. For instance, other known ligands of PLXNB2, such as SEMA5A and SEMA4D, are either not expressed or not detectable in breast tumors at the primary site. However, the interaction of SEMA4D in disseminating breast cancer cells with PLXNB1 of brain endothelial cells can contribute to the brain-tropic metastasis of cancer[89]. One of the future studies is to examine the association of PLXNB2 with CNS diseases and its contribution to regulating metastasis of breast cancer to the brain and other organs.

A recent study by Borrelli et al. has identified PLXNB2 as a crucial host-derived regulator of liver colonization in colorectal and pan-creatic cancer through its interactions with class IV semaphorins on tumor cells, leading to the upregulation of KLF4 and promoting the acquisition of epithelial traits[86,86]. These findings suggest that PLXNB2-semaphorin signaling plays a pivotal role in tumor metastasis across different cancer types, making it a promising therapeutic target for further investigation. Furthermore, SEMA4D interacts with PLXNB2 to mediate monocyte adhesion to endothelial cells[90]. This interaction is significant because it suggests that PLXNB2 not only plays a role in tumor cell behavior but also influences immune cell recruitment and adhesion, which can impact the tumor microenvironment and meta-static niche. In cancer, this could mean that PLXNB2-semaphorin sig-naling may contribute to forming a more adhesive and supportive environment for both cancer cells and immune cells, facilitating metastasis. Understanding the dual role of PLXNB2 in both tumor cell adhesion and immune cell dynamics could offer new therapeutic

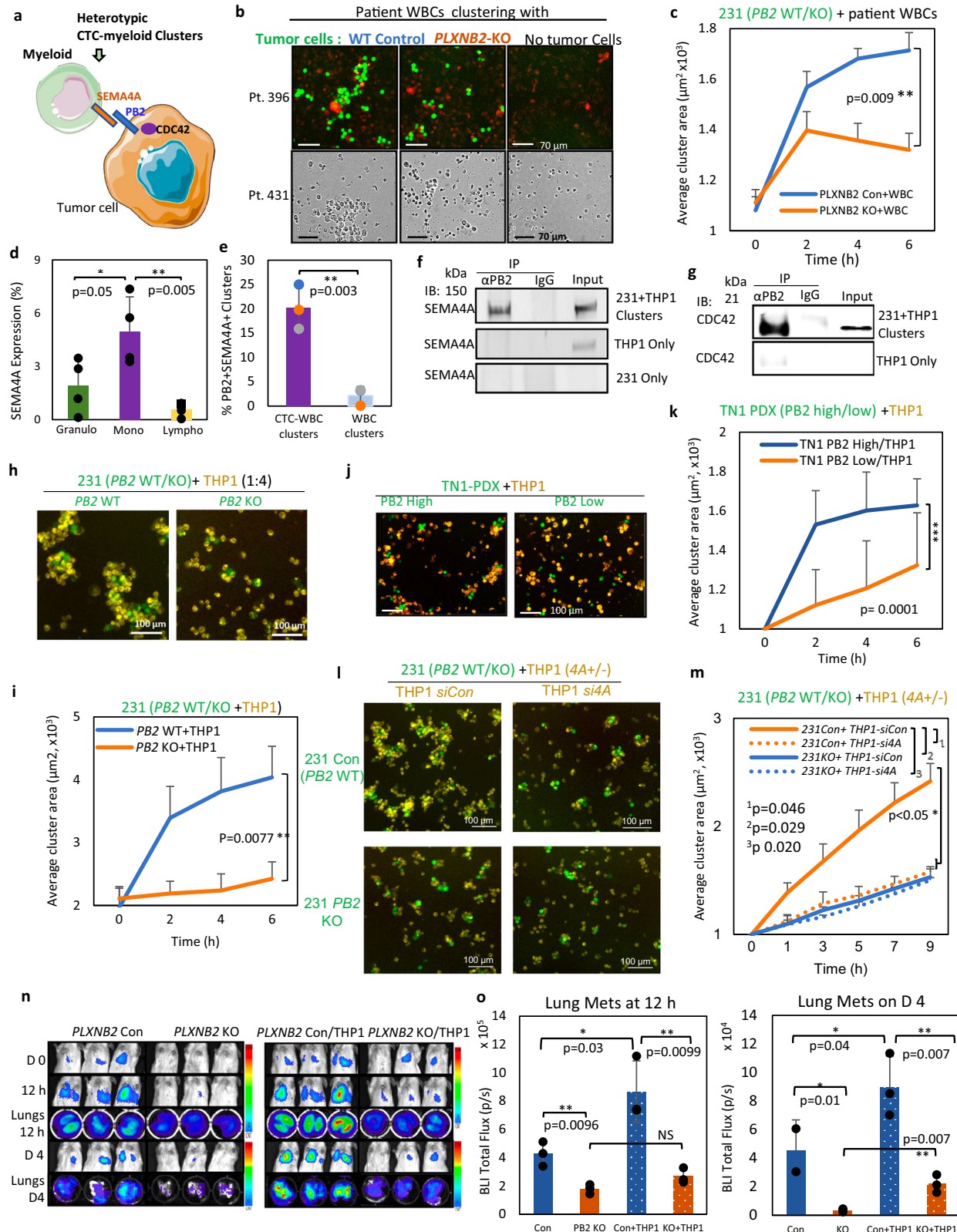

strategies aimed at disrupting these processes to reduce metastasis and improve patient outcomes.

Ongoing studies remain necessary to overcome the limitations of our study. First, the RScore approach for ranking candidate proteins is a semi-quantitative, heuristic-based method that integrates multiple modalities of biological datasets and clinical relevance. It is designed with a modifiable constant factor (weight) for each data input to

provide room and flexibility for continuous training and machine learning (ML)-based determination in following studies. It will be optimized through integration with expanded dataset inputs, both known and unknown, and experimental validation of candidate rankings to enhance predictive power. Second, the use of fluorescence-activated cell sorting (FACS) for analyses of CTCs and CTC clusters requires cross validation via other methods, including but not limited

**Fig. 5 | PLXNB2 binding to SEMA4A on monocytes drives heterotypic clustering. a** Schematic of the interactions between breast cancer cell PLXNB2 and monocyte SEMA4A for heterotypic cluster formation. **b** Representative images of heterotypic clusters of L2G+ MDA-MB-231 control or *PLXNB2* KO cells with WBCs from breast cancer patients (Pt). Top: fluorescent images with WBCs from Pt 396 (minimal Cytotox red-stained dead cells). Bottom: brightfield images with WBCs from Pt 431. An average of 13581 and 11376 WBCs is associated with Con and KO clusters, respectively. **c** Heterotypic cluster area curves as shown in (**b**) ($n = 5$ replicates examined over 4 individual experiments). P-value was calculated using two-sided unpaired t-test. **d** SEMA4A expression on granulocytes, monocytes, and lymphocytes from four patients with advanced breast cancer. P-values were calculated using one-sided ANOVA. **e** Percent of heterotypic CTC-WBC clusters vs. WBC clusters that express both PB2 and SEMA4A, $n = 4$ patient samples. P-value was calculated using two-sided unpaired t-test. **f, g** Immunoblots of SEMA4A (**f**) and CDC42 (**g**) with cell lysates immunoprecipitated with anti-PlexinB2 (αPB2) or IgG isotype. **h, i** Representative images (**h**) and cluster area curves (**i**) of heterotypic

clustering of MDA-MB-231 cells (green) and THP1 monocytes (yellow) mixed at a 1:4 ratio, $n = 3$ technical replicates examined over 3 individual experiments. P-value was calculated using two-sided unpaired t-test. **j, k** Representative images (**j**) and cluster size curves (**k**) of heterotypic clustering with L2G+ TN1 PDX tumor cells (green, sorted by PLXNB2 high and low) and THP1 (yellow), $n = 3$ technical replicates examined over 3 individual experiments. P-value was calculated using two-sided unpaired t-test. l-m) Representative images (**l**) and cluster size curves (**m**) of MDA-MB-231 Con or KO cells clustering with THP1 (transfected with *siCon* or *siSEMA4A*), $n = 3$ technical replicates examined over 3 individual experiments. *P* values were calculated using one-sided ANOVA. n) BLI images of NSG mice after tail vein injection of pre-clustered MDA-MB-231 cells alone (*PLXNB2* Con or KO), and with THP1 cells (*PLXNB2* Con/THP1, KO/THP1), $n = 3$ mice/group. *P* values were calculated using one-sided ANOVA. **o** BLI flux of lungs, $n = 3$ mice/group at each time point. Data are presented as mean values ± SD. Source data are provided as a Source Data file.

to multiplexing fluorescence imaging and/or single-cell sequencing of enriched CTCs from fixed or unfixed blood cells ex vivo (i.e., Cell-Search, Parsortix, ImageStream, CellView, etc) as well as intra-vascular CTCs in tissue sections in situ (no blood processing).

## Methods

### Patient sample collections
Blood samples and breast tissue sections (tumors and normal adjacent) were collected from stage III-IV breast cancer patients under guidelines from the Institutional Review Board and Ethics committee at Northwestern University (IRB protocols STU00203283 and STU00214936) in compliance with NIH human subject studies guidelines. Blood samples were collected into CellSave tubes for CellSearch analyses or into EDTA tubes for flow cytometry of live cells. The Cell-Search® platform for CTC detection (CD45⁻DAPI+Cytokeratin+) had one open immunofluorescence channel for Plexin B2 analysis. Prior to flow cytometry analysis, blood specimens in EDTA tubes underwent red blood cell lysis (Sigma #R7757). Breast tumor tissues and normal adjacent tissues were frozen in tissue freezing buffer OCT until sectioned for laser capture microdissection (LCM) for mass spectrometry analysis.

### Animal studies
All animal procedures complied with the NIH Guidelines for the Care and Use of Laboratory Animals and were approved by the Northwestern University Institutional Animal Care and Use Committee (IACUC protocol IS00014098). The maximal tumor size/burden (2 cm in diameter) was not exceeded in any case. All mice used in this study were kept in specific pathogen-free facilities in the Animal Resources Center at Northwestern University.

### CellSearch
CellSearch analysis processed 7.5 mL of patient blood using the CTC Epithelial Kit (CellSearch #7900001) and CXC Kit (CellSearch #7900017) to deplete immune cells by an EpCAM+ selection and identify specific markers. Cells were then stained for CK, DAPI, and CD45 (3 mL for each, Menarini # 7900001) and Plexin B2 (1:1000, Miltenyl Biotec #130-126-566).

### Flow cytometry analysis
Mouse/human cells, PDXs, and patient cells were counted and resuspended in wash buffer (PBS + 2% FBS). They were blocked with mouse IgG (Sigma #I5381) for 10 minutes on ice and incubated with fluorescence-conjugated antibodies for 15 minutes on ice: Plexin B2 PE (phycoerythrin) (1:1,000, human, R&D Systems #FAB53291P), Plexin B2 APC (1:1,000, human, R&D Systems #FAB53291A), Plexin B2 FITC (1:1,000, mouse, R&D Systems #FAB6836G), CD45 (1:1,000, human, BD Bioscience #557748), EpCAM FITC (1:1,000, human, BD Bioscience

#347197), SEMA4A PE (1:1,000, human/mouse, BioLegend #148404), SEMA4A APC (1:1,000, human/mouse Biolegend #148406). Cells were washed 2x in wash buffer and run for analysis on a fluorescence-associated cell sorting (FACS) LSR cytometer from BD Biosciences.

### Cell sorting
Human/PDX/PBMC cells were resuspended in PBS + 2% FBS at a final concentration of $10 \times 10^6$ cells/mL in 5 mL FACS tubes after filtering (Fisher Scientific #352235). Cells were blocked with mouse IgG (Sigma #I5381) for 10 minutes on ice and florescent antibody for 15 minutes on ice: Plexin B2 PE (1:1000, human, R&D Systems #FAB53291P), Plexin B2 APC (1:1000, human, R&D Systems #FAB53291A), Plexin B2 FITC (1:1000, mouse, R&D Systems #FAB6836G), SEMA4A PE (1:1000, human/mouse, BioLegend #148404). Cells were then run through BD FACSMelody Cell Sorter and cells were collected based on gated populations. Collected cells were washed 2x in PBS prior to downstream application.

### Tissue and cell preparation for mass spectrometry analysis
Human breast cancer tissues were frozen with OCT and stored at −80 °C until being cryo-sectioned to a series of 10-μm thick sections using a blade temperature −35 °C and specimen temperature of −25 °C. The sections were then thawed and mounted onto PEN membrane slides for laser capture microdissection (LCM) and onto regular glass slides for H&E staining or immune-histochemistry staining. For LCM, the OCT was removed by immersing slides into 50% ethanol for 2 min, dehydration in100% ethanol (2 min) and xylenes (1 min), and then the sections dried out in a vacuum desiccator. After identified via H&E staining of adjacent sections, and the regions of tumor and non-tumor/normal adjacent were correspondingly identified in the unstained tissue sections for LCM on a PALM MicroBeam system (Carl Zeiss MicroImaging, Munich, Germany), at an energy level of 63 and with an iteration cycle of 1. Voxels were collected within a wet buffer droplet for the best protein recovery.

MDA-MB-231 breast cancer tumor cells underwent double transfection of *siPB2* for KD of *PB2* (Dharmacon #L-031513-01) and control siRNAs (Dharmacon #D-001810-10-50). Cells were then trypsinized and resuspended on PolyHema-coated plates and allowed to cluster for 4 h. Adherent cells were scraped from the plate at the zero hour time point. After clustering, cells were harvested and lysed using RIPA buffer (VWR Amresco #N653-100mL) supplemented 1:100 with protease inhibitor (Thermo Fisher #78440).

### Protein digestion by S-Trap for proteomic analysis
Tissue voxels were collected in digested 10 μL of a cocktail of buffer containing 0.1% DDM in 50 mM TEABC, 20 ng of Lys-C, and 80 ng of trypsin. MDA-MB-231 cell lysates were resuspended in SDS buffer (the

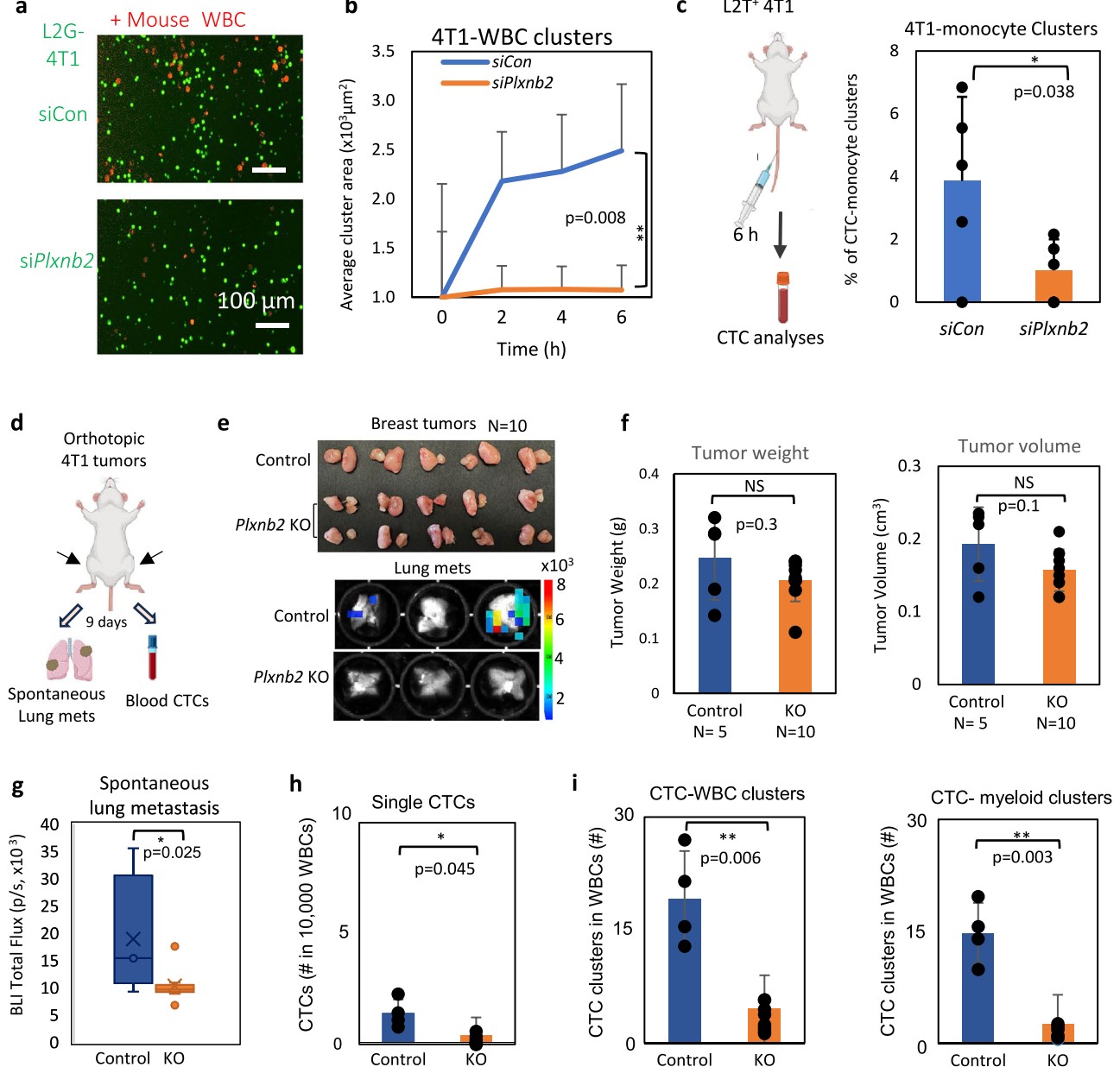

**Fig. 6 | Mouse *Plxnb2* depletion inhibits CTC clustering and spontaneous metastasis of 4T1 tumors in vivo. a**, **b** Representative images at 6 h clustering (**a**) and quantification (**b**) of 4T1 cells transfected with *siCon* and *siPlxnb2* during co-clustering with mouse white blood cells over 6 h; *n* = 5 technical replicates examined over 3 individual experiments, *p* = 0.008. **c** Schematic and quantification of % of heterotypic mouse 4T1 CTC clusters with monocytes at 6 h after tail vein injection of the tumor cells transfected with siRNA control (Con) and si*Plxnb2* (>80% knockdown efficiencies by flow). $5 \times 10^5$ 4T1 tumor cells were injected into Balb-c mice via the tail vein and cardiac blood was collected at 6 h for CTC analysis via flow cytometry (*n* = 5 mice/group). **d–i** Schematic of orthotopic 4T1 breast tumor implantations ($1.5 \times 10^6$ cells) at the L4/R4 mammary fat pads and following metastasis and CTC analyses (**d**), representative photos of 4T1 tumors (Control and *Plxnb2* KO) and ex vivo lung bioluminescence images on Day 9 (**e**), and quantified tumor weight and volume (**f**), lung metastasis (total flux of bioluminescence) (**g**) and CTCs, including single CTCs in live blood cells (**h**) as well as 4T1-WBC heterotypic clusters and CTC-myeloid clusters (**i**) among all white blood cells (WBCs), shown as # of events in 10,000 WBCs, *n* = 5 mice/group. Tumor burden includes both L4/R4 tumors in each mouse and is used to normalize lung metastasis signals in (**g**). Data are presented as mean values +/-SD, with *P* values reported from two-sided unpaired t-tests. Source data are provided as a Source Data file.

final concentration was 5%). The protein concentration was measured with a BCA protein assay (Thermo Fisher Scientific). A total of 50 mg of protein was reduced with 10 mM DTT for 15 minutes at 37 °C and subsequently alkylated with 50 mM iodoacetamide at 25 °C for 15 minutes in the dark. The samples were acidified by phosphoric acid (the final concentration was 2.5%) and then diluted with six volumes of "binding" buffer (90% methanol; 100 mM triethylammonium bicarbonate, TEAB; pH 7.1). After mixing, the protein solution was loaded onto an S-Trap filter from Protifi (Huntington, NY), spun at 10,000 g for 1 minute, and then the filter was washed with 150 μL of binding buffer three times. Proteins were digested with Lys-C (Wako) and sequencing-grade trypsin (Promega, V5117) (1 μg of each in 20 μL of 50 mM TEAB) in the filter at 37 °C for 16 h. To elute peptides, 40 μL of 50 mM TEAB, 40 μL of 0.2% formic acid (FA) in $H_2O$, and 40 μL of 50% acetonitrile (CAN) in $H_2O$ were added sequentially. The peptide solutions were pooled for BCA assay to estimate peptide amounts. Totals

of 20 μg of peptides were dried with a SpeedVac and stored at −80 °C until LC-MS/MS analysis.

LC-MS/MS analysis of tumor tissue voxels, patient CTCs, CTC-derived PDXs, and MDA-MB-231 cells: Lyophilized peptides were reconstituted in 200 μL of 0.1% TFA with 2% ACN containing 0.01% n-Dodecyl $\beta$-D-maltoside (DDM)[91] to reach a concentration of 0.1 μg/μL, and 5 μL of the resulting sample was analyzed by LC-MS/MS using an Orbitrap Fusion Lumos Tribrid mass spectrometer (Thermo Scientific) connected to a nanoACQUITY UPLC system (Waters) (buffer A: 0.1% FA with 3% ACN and buffer B: 0.1% FA in 90% ACN) as previously described[92]. Peptides were separated by a gradient mixture with an analytical column (75 μm i.d. × 20 cm) packed using 1.9-μm ReproSil C18 and with a column heater set at 50 °C using an LC gradient of 2–6% buffer B in 1 min, 6-30% buffer B in 84 min, 30-60% buffer B in 9 min, 60-90% buffer B in 1 min, and finally 90% buffer B for 5 min at 200 nL/min. The DIA-MS/MS scan was performed in the HCD mode with the following parameters: precursor ions from 350–1650 *m/z* were scanned at 120,000 resolution with an ion injection time of 60 ms and an AGC target of 1E6. The range of *m/z* (isolation window) of data-independent acquisition (DIA) windows from 377 (54), 419 (32), 448 (28), 473.5 (25), 497.5 (25), 520.5 (23), 542.5 (23), 564.5 (23), 587 (24), 610.5 (25), 635 (26), 660 (26), 685.5 (27), 712.5 (29), 741 (30), 771 (32), 803.5 (35), 838.5 (37), 877 (42), 921 (48), 972 (52), 1034.5 (71), 1133.5 (129), and 1423.5 (453) was scanned at 30,000 resolution with an ion injection time of 120 ms and an AGC target of 3E6. The isolated ions were fragmented with HCD at the 30% level.

## Proteomic data analysis

The raw DIA data were processed with Fragpipe[93,94] and searched against a human UniProt database (FASTA file dated Jan. 05, 2022 with 40,818 sequences, which contained 20,409 decoys). Initial fragment mass tolerances were set to 20 ppm. A peptide search was performed with strict tryptic digestion (Trypsin) and allowed a maximum of two missed cleavages. Carbamidomethyl (C) was set as a fixed modification; acetylation (protein N-term) and oxidation (M) were set as variable modifications. DIA_SpecLib_Quant workflow was used for DIA quantitation (takes DIA data as input, builds a spectral library using MSFragger-DIA, then quantifies with DIA-NN).

Integrated ranking of proteins based on their relative abundance, tumor specificity, and significance in association with patient outcomes: The presence of proteins in chosen datasets was first determined. The integrated ranking of each protein is formulated as Rscore $= \sum n_{i=1}(r_i c_i)/n$ which sums each protein's individual ranks $(r_i)$ multiplied with a constant factor $(c_i)$ as its weight factor and is divided by n (number of integrated ranks). The individual ranks in this study include (but not limited to):

$$R_{score} = \frac{r_1 c_1 + r_2 c_2 + r_3 c_3 \ldots + r_n c_n}{n} \quad c_i = constant\,factor\,(weight\,of\,each\,rank) \tag{1}$$

$$r_i(pa) = rank\left(\frac{protein\,abundance/intensity\,in\,MS\,spectral\,counting}{candidate\,protein\,length(a.a.)/median\,pr.length(a.a.)}\right) \tag{2}$$

$$r_i(sc) = rank(significant\,changes : p, ratio\,or\,fold\,change, absolute\,change) \tag{3}$$

$$r_i(ca) = rank(clinical\,association : p, hazard\,ratio) \tag{4}$$

(1) $r_i$ (pa), ranks in relative protein abundance in multiple mass spectrometry (MS) datasets (breast tumors, cancer cell lines, and CTCs), calculated as protein intensity (total number of MS/MS spectra for a given protein) divided by the ratio of the tested protein length (# of amino acids) versus the median protein length (# of amino acids).

The ranks of detected proteins (x) in a dataset ranged from the highest abundance ($r_i = 1$) to the lowest ($r_i = x$).

(2) $r_i$ (sc), ranks in significant changes (p values, ratio or fold change, and absolute change), such as tumor-specific expression of a chosen protein in TNBC voxels via laser capture microdissection vs normal adjacent tissues. The ranking scored from the lowest to highest in p values, from the highest to lowest in fold changes (ratios), and from the highest to lowest in absolute differences.

(3) $r_i$ (ca), ranks in clinical association (p values, hazard ratios) of each protein with overall survival (OS) and distant metastasis-free survival (DMFS) among multiple datasets. The log rank P values are calculated based on the most significant cut-off values on multiple Cox regression tests to separate two groups of high and low expression as cited[35] (PMID: 34309564) unless specified. The ranking for oncogenic proteins scored from the lowest to highest in log rank p values (negative association) and from the highest to lowest in the hazard ratios.

The ranks of detected proteins (x) in a dataset ranged from the highest rank ($r_i = 1$) and to the lowest ($r_i = x$). The constant factor ($c_i$) of each individual rank reflects its contribution weight and can be determined by computational estimation and experimental analyses. The top R scores are of the smallest sum values. Lab-obtained proteomic datasets as well as published datasets were utilized for integrated analyses of adhesion proteins in the study.

Adhesion proteins (N = 608, derived from the Molecular Signature Database[27–29] of the Gene Ontology Biological Processes[30,31]) were ranked according to tumor specific expression in the global MS proteomic datasets of laser capture microdissection voxels of human TNBC regions versus normal adjacent tissues (p values, ratio or fold change, absolute differences), relative protein abundances in 122 treatment-naive primary breast tumors of patient samples[32], patient CTCs, patient-derived xenograft (PDXs), and TNBC cells such as MDA-MB-231[33] and Hs578T cells[34]. Relative protein abundance in other datasets such as THP1 macrophage cells[95] and expression in extracellular vesicles (EVs) was also ranked. Patient outcome association analyses were conducted via KMplotter data in individual datasets (protein and mRNA expression) and ProteinAtlas data (mRNA expression). The p values and hazard ratios of each protein in negative or positive OS and DMFS were ranked (see Supplementary Data 1).

## Extracellular vesicle isolation

EVs were isolated from human cell lines and PDX models from culture media after 72 h in culture. Media were then pooled and isolated using differential centrifugation. The first centrifugation step was at 2000 x g for 10 minutes. Media were then transferred into an appropriate ultracentrifuge tube (Beckman Coulter # 344058) and spun at 10,000 x g for 30 minutes. Supernatant was removed and transferred into a clean ultracentrifuge tube and spun at 100,000 x g for 70 minutes. The resulting EV pellet was washed with PBS and spun again at 100,000 x g for 70 minutes. The final EV pellet was resuspended in PBS and stored at −80 °C until analysis.

## Cell culture

All cell lines used in this study were obtained from ATCC and periodically tested to be mycoplasma-negative using Lonza's MycoAlert Mycoplasma Detection Kit (cat #LT07-218). Cell lines were maintained in complete supplemented media (10% FBS, 1% penicillin-streptomycin (Sigma-Aldrich P4333-100ML)). MDA-MB-231 and 4T1 cell lines were maintained in DMEM supplemented with high glucose (Corning #10-013-CV) for <20 passages in cell culture incubators at 37 °C, 5% $CO_2$. MDA-MB-468, HS578T, and THP1 cells were cultured in RPMI-1640 supplemented media (Fisher Scientific #SH30027.01) for <20 passages in cell culture incubators at 37 °C, 5% $CO_2$. No commonly misidentified cell lines were used in the study.

For tumor cell clustering analyses, we used multiple human breast cancer cell models, including HER2 + SKBR3 and TNBC MDA-MB-231 cells. MDA-MB-231 cells not only form tumor cell-tumor cell clusters, but also adhere with human WBCs and monocytes (THP1 cells) to form heterotypic clusters. In addition, mouse breast cancer line 4T1 has shown similar phenotypes as MDA-MB-231 cells. Therefore, we focused on using MDA-MB-231 and 4T1 for functional studies in vivo. For tail vein injection and spontaneous metastasis studies, we used MDA-MB-231 cells (WT or *PLXNB2* KO) in female NSG mice (6–8 weeks old). For mouse *Plxnb2* KO effect in lung metastasis, we used 4T1 cells (WT or *Plxnb2* KO) in female Balb/c mice (6–8 weeks old).

### Gene knockdown

A total of $2 \times 10^6$ cells were plated in a 10 cm dishes one day prior to KD (day 0). On the day of KD, ON-TARGETplus siRNA at a final concentration of 50 nM per plate was incubated with Dharmafect 1 at 100 nmol/L (GE Dharmacon #T-2001-03) for 20 minutes at room temperature in reduced serum Opti-MEM Medium (Thermo Fisher Scientific #31985070). siRNA+Dharmafect mixture was added to cells in Opti-MEM for 16-24 h. Cells were passaged in complete media and reseeded at $2 \times 10^6$ cells in a 10 cm dish, and the KD procedure was repeated once more as described. On day 4, cells were harvested, counted, and analyzed via flow cytometry and western blotting to check for sufficient KD of target protein. The following siRNAs were used: SMARTpool Human *PLXNB2* (Dharmacon #L-031513-01), SMARTpool Mouse *Plxnb2* (Dharmacon #L-040980-00-0010), SMARTpool Human *SEMA4C* (Dharmacon #L-015364-01-0010), ON-TARGETplus *PLXNB2* Human siRNA-09 (Dharmacon #J-031513-09-0010), ON-TARGETplus *PLXNB2* Human Individual siRNA-10 (Dharmacon #J-031513-10-0010), ON-TARGETplus *PLXNB2* Human Individual siRNA-11 (Dharmacon #J-031513-11-0005), ON-TARGETplus *PLXNB2* Human Individual siRNA-12 (Dharmacon #J-031513-12-0005), ON-TARGETplus Non-targeting Pool (Dharmacon #D-001810-10-50).

### Gene overexpression

Cells were transfected with *PLXNB2* full-length and mutant overexpression plasmids via a Lipofectamine 3000 Transfection Kit (Thermo Fisher #L3000015). Cells were plated at 300k cells/well in a 6-well plate the day prior to transfection. After transfection, cells were incubated for 48-72 h under standard growth conditions and then harvested to check expression of target protein via flow cytometry and western blotting. The following overexpression vectors were used: pLV-*PLXNB2*-mRBD (Addgene #86240), pLV-*PLXNB2*-dVTDL (Addgene 86239), pLV-*PLXNB2*-dECTO (Addgene #86238), *PLXNB2* OHu01778C_pcDNA3.1(+) N-Terminal Flag-Tag (GenScript #SC1626), *PLXNB2* OHu01778D_pcDNA3.1 + /C- C-Terminal Flag-Tag (GeneScript #OHu01778D), *PLXNB2* Untagged Construct (GeneScript #SC1625).

### Gene knockout

Two individual pre-designed human *PLXNB2* sgRNA CRISPR-Sanger clones and one non-targeting control vector (Sigma-Aldrich #0020) were ordered from Sigma as glycerol stock (Sigma-Aldrich #HS5000013567, #HS500013568). Bacteria were expanded for maxi prep according to kit protocols (Qiagen #12163). Plasmid was isolated and used to create lentivirus. Cells were infected with either of the sg*PLXNB2* clones or the control sgRNA with Cas9-GFP virus (Sigma-Aldrich #0030) at 10 IFU/cell in Opti-MEM (Thermo Fisher Scientific #31985070) with 8 μg/mL supplemented Polybrene (Millipore Sigma #TR-1003-G). Cells were incubated for 4 h at 37 °C in 5% $CO_2$ upon which complete medium was added to the culture. Cells were incubated in normal growth conditions for 48-72 h and monitored for GFP expression. Cells were harvested and analyzed for sufficient KO of protein. KO cells were sorted for multiple pooled clones using the BD FACSMelody Automated Cell Sorter based on PLXNB2 expression.

For mouse *Plxnb2* KO generation, we transfected two individual TrueGuide Synthetic gRNAs (CRISPR66859_SGM and CRISPR66855_SGM, Thermo Fisher) into Cas9-expressing 4T1 cells. The gRNAs were resuspended in 100 μM stock in TE buffer. One day before transfection, cells were seeded in a 6-well plate at 30% confluency. For each well, 37.5 pmol of gRNA was added to 125 μL Opti-MEM I Medium, and then 7.5 μL of Lipofectamine CRISPRMAX Cas9 Transfection Reagent was added to the Medium. The gRNA and Lipofectamine tubes were incubated for 5 minutes at RT, and then combined and incubated for 10 minutes at RT. Transfection complex was added to the wells, and cells were incubated at 37 °C for 2 days, and the *Plxnb2* KO was validated by flow cytometry.

### Clustering assay

In homotypic tumor cell clustering assays, 20,000 tumor cells were plated in Poly (2-hydroxyethyl methacrylate) (PolyHema)-coated 96-well flat bottom plates (Sigma-Aldrich #P3932-25G). Tumor cell lines were trypsinized from the plate and resuspended by pipetting up and down 10 times and lightly vortexing the samples for 5 seconds immediately before plating. This procedure was optimized by cell imaging under a microscope and/or in the Incucyte at Time 0 h to ensure cells were in a single cell suspension. Tumor cells from PDXs were dissociated from primary tumors (maintained in NSG mice) and seeded on collagen-coated plates for clustering (as terminal experiments) based on established protocols[16,96].

In heterotypic tumor cell-immune cell clustering assays, tumor cells and immune cells were plated at a 1:4 ratio in PolyHema-coated 96-well flat-bottom plates. Cells were monitored in the Incucyte Live Cell Imager (Sartorius) for 24 h and analyzed for average cluster size over time, with a cluster being defined as two or more cells.

Cluster assay analysis was performed using the built-in image analysis tool of IncuCyte software, based on pre-defined masks and cluster objects. The processing definition applied to cluster assays depends on the cell type used to account for individual differences in the cell size. Clusters in breast cancer cell lines, for example, are defined by their surface area from a minimum of 2 cells (object area >300-500 μm$^2$). The pooled analyses of clusters are based on the calculated area of tumor cells and cluster counts. To help visualize the clustering phenotype, a cell cluster of approximately 2-3 cells are in 500 μm$^2$, 4-5 cells in 1,000 μm$^2$ and 6-12 cells in 1,500-3,000 μm$^2$ (PDX models), 2 cells in 750 μm$^2$, 4 cells in 1,500 μm$^2$, and 10-12 cells in 3,500 μm$^2$ (cell line models).

### Immunoblotting (western blotting)

Cells were pelleted and then lysed in 1x RIPA Lysis Buffer (VWR Amresco #N653-100mL) supplemented with 1:100 protease inhibitor cocktail (Thermo Fisher #78440) and incubated on ice for 30 minutes. Lysates were then centrifuged for 10 minutes at 14,600 x g at 4 °C. Cell lysates were measured for protein concentration using Bradford analysis (Thermo Fisher #23209, Bio-Rad #500-0006). Totals of 10-30 μg of protein lysate and 10 μL of dual color protein standard ladder (Bio-Rad #161-0374) were loaded onto 4-20% Mini-PROTEAN gels (Bio-Rad #4568094) and then transferred to nitrocellulose membranes using the Bio-Rad TurboTransfer system and kits (Bio-Rad #1704270). Membranes were blocked with 5% milk in Tris-buffered saline and 0.1% Tween 20 (TBST) for 60 minutes and then washed in 0.1% TBST. Primary antibody was incubated on membranes in 5% milk in TBST for 1.5 h at room temperature or at 4 °C overnight. Secondary HRP-conjugated antibodies were added at a dilution of 1:10,000 in 5% milk in TBST and incubated for 60 minutes at room temperature (Anti-Mouse: Promega #W402B; Anti-Rabbit: Promega #W401B). The Pierce SuperSignal West Pico PLUS chemiluminescent substrate (Thermo Scientific #34577) was added one minute prior to imaging using the Bio-Rad Chemidoc imaging system. Primary antibodies used include: Anti-Plexin B2 (1:500, Protein Tech #10602-1-AP), Anti-β-Actin (1:1,000,

Abcam #AB8224), Anti-SEMA4A (1:1,000, Thermo Fisher #PA5-101258), Anti-SEMA4C (1:1,000, Ray Biotech #102-11819), Anti-SEMA4G (1:1,000, Santa Cruz Biotech #sc-515644), Anti-SEMA4D/CD100 (1:1,000, Santa Cruz Biotech #sc-39065), Anti-SEMA7C (1:1,000, Santa Cruz Biotech #sc-376149).

## Co-immunoprecipitation

The co-immunoprecipitation protocol was done as directed in the Dynabeads Co-Immunoprecipitation Kit (Thermo Scientific #14321D). Plexin B2 (Protein Tech #10602-1-AP) and control rabbit IgG (Protein Tech #3000-0-AP) were pre-conjugated to Dynabeads at 7 µg antibody/mL of beads. Cells were lysed using immunoprecipitation-lysis buffer and incubated overnight at 4 °C with 7.5 mg of pre-conjugated Dynabeads for every 1-15 g protein. Beads were washed, and protein was eluted using SDS loading buffer for downstream applications of western blotting and mass spectrometry analysis.

## PDX mouse models

Multiple PDX models of human breast cancer were previously established in the lab[13,16]. Cells from TNBC patient tumors or pleural effusion were used to establish tumors that propagated in immunodeficient NSG mice. PDX models were labeled with either Luc2-tdTomato (L2T) (red) or Luc2-eGFP (L2G) (green) reporters to track tumor growth and measure metastasis using bioluminescence imaging and fluorescence analyses (microscopy or flow cytometry). Models were maintained in the lab through tumor harvesting, dissociation, and re-implantation of tumor cells into the mammary fat pads of female NSG mice (6-8 weeks old). PDX models were routinely checked for expression/lack of expression of key markers to monitor phenotype. PDX models were sorted for PB2^high/low cells for downstream in vitro experiments and analysis.

## Lung colonization via tail vein injection and bioluminescence imaging

All mice used in this study were female NSG mice 6–12 weeks of age (Jackson Laboratory) and housed in the pathogen-free barrier facility in the Animal Resources Facility at Northwestern University. All animal studies were completed under approved protocols (IS00014098) and adhered to all procedures and regulations outlined by the NIH Guidelines for the Care and Use of Laboratory Animals. Animals were randomized by age and weight and excluded from experiments if presenting conditions unrelated to tumors. Mice were injected with 200,000 L2G or L2T-labeled MDA-MB-231 cells (WT or *PLXNB2* KO) via the tail vein, and lung colonization was monitored upon intraperitoneal injection of luciferin (Gold Bio #LUCK-1G 115144-35-9). The bioluminescence signals of metastasis burden were imaged with the IVIS Spectrum Imager, using same imaging time (acquisition times ranged from 5 s to 5 min) across all groups with identical region of interest quantified for comparison. Mice were kept for four days post-injection or until the survival endpoint. At the experimental endpoint, lungs were harvested, imaged in PBS ex vivo for bioluminescence on black-wall 24 well plates, and then preserved in formalin (Fisher Scientific #SF98-4) for H & E staining. The bioluminescence imaging (BLI) signals for regions of interest were measured as total flux (photons/second, p/s). When applicable, the tumor growth signals were normalized and presented as fold change in comparison to that of Day 0. To assess spontaneous lung metastasis, the BLI signals (total flux, p/s) were normalized by tumor weight to minimize the effects of tumor burden.

## Spontaneous metastasis in vivo

Female NSG mice at the age of 6–8 weeks were injected with 10,000 L2T/L2G-labeled human breast tumor MDA-MB-231 cells (WT or *PLXNB2* KO) into each lower mammary fat pad (L4 and R4) for 6–10 weeks of monitoring. For mouse L2T/eGFP-labeled 4T1 models, female Balb/c mice at age of 6–8 weeks received $1.5 - 2.0 \times 10^6$ tumor cells per mammary fat pad (L4 and R4) and monitored for 9 days before harvest. Tumors were monitored at least weekly for growth, and lung metastasis was monitored using bioluminescence imaging. Human tumors grew for 8 weeks and mouse 4T1 grew for 9 days (prior to anti-L2T/eGFP immunity development) or until survival endpoint, at which point tumors, lungs, and blood were collected and analyzed ex vivo. Lungs were imaged under bioluminescence and microscopy and preserved in formalin (Fisher Scientific #SF98-4) for hematoxylin-eosin (H&E) staining and immunohistochemistry (IHC) analysis. Tumors were weighed and preserved in formalin for H&E/IHC analysis. Blood was collected, and red blood cells were lysed using RBC lysis buffer (Sigma #R7757) followed by analysis via flow cytometry for L2G/L2T-positive tumor cells.

## Tumorigenesis (Limiting dilution assay)

Serial dilutions of cells were prepared to achieve 1000, 100, and 10 cells per injection in a total volume of 100 µL mixture consisting of a 1:1 mixture of Matrigel and PBS. Female NSG mice (6–8 weeks old) were injected with L2G-labeled human breast tumor MDA-MB-231 cells (WT or *PLXNB2* KO) into the mammary fat pads. Each dilution group included three mice, and each mouse received injections of both WT and *PLXNB2* KO cells from same dilutions, two injection per flank. Mice were monitored for tumor formation over 6 weeks. Tumor growth was assessed twice weekly by measuring bioluminescence signals using the IVIS Spectrum Imager with consistent imaging acquisition times (5 min per session).The frequency of tumor-initiating cells (TICs) was calculated using extreme limiting dilution analysis (ELDA) software (http://bioinf.wehi.edu.au/software/elda/).

## Orthotopic injection and cell cycle analysis of CTCs

L2G- or L2T-labeled MDA-MB-231 cells (WT and *PLXNB2* KO) were mixed in equal numbers, with a total of $2 \times 10^5$ cells resuspended in a 1:1 mixture of Matrigel and PBS (50 µL each). The cell suspension was injected into both lower mammary fat pads of each mouse. Each group included five female NSG mice (6–8 weeks old). Tumor growth was monitored for 8–10 weeks. At the endpoint, tumors were dissected and weighed, and lung metastases were imaged using bioluminescence with the IVIS Spectrum Imager. Blood samples were processed with red blood cell lysis buffer and stained with Hoechst dye (Thermo Fisher, Cat# 33342) diluted 1:1000 for cell cycle analysis of CTCs. CTCs were identified based on their L2G or L2T fluorescence signals.

## Tumor microarray (TMA) and IHC

A total of 89 formalin-fixed paraffin-embedded breast tumor tissues were included in the tumor TMA, with selected tumor regions guided by H&E-stained images. To make a TMA that allows microscopic comparison of the staining characteristics of different blocks while preventing exhaustion of pathological material, a core of paraffin was removed from a "recipient" paraffin block (one without embedded tissue), and the remaining empty space was filled with a core of paraffin-embedded tissue from a "donor" block. An H&E-stained recipient block that is representative of the tissue remaining in the donor block was used to select the sample core with a color marker corresponding to tumor, benign, etc. Matched blocks were pulled out and a recipient TMA block was made and trimmed with the face of the block even with the size of a 1.5 mm core by using the semi-automatic Veridiam Tissue Microarrayer VTA-100. The created TMA block was sectioned for staining. In this TMA, 19 cases from the STU00203283, 9 ER-negative cases, 30 triple-negative cases, 27 ER-positive cases, and 4 normal breast cases (see Supplementary Data 3) were selected and used to construct the recipient block. Detailed patient information can be found in Supplementary Data table. IHC was performed with the help of Bella Shmaltsuyeva of the Robert H. Lurie Comprehensive

Cancer Center Pathology Core Facility with Plexin B2 antibody (1:500, Protein Tech #10602-1-AP).

The definition of PlexinB2 high (positive) and low (weak positive and negative) in the TMA analyses followed pathologist recommendations that apply to IHC analyses with both protein staining intensity (positive vs negative) and areas (%): any sample with > 10% positive tumor regions was called positive (high), anything between 1-10% was called low positive and anything <1% was called negative. The relatively low intensity of overall PlexinB2 staining as shown in Supplementary Fig. S2a (low control) was also considered weak positive and included in the PlexinB2 low group.

## H&E staining

Mouse lungs and tumors were paraffin-embedded, and sections were mounted on slides and processed with H&E staining with help from the Mouse Histology and Phenotyping Laboratory at Northwestern University.

## Mammosphere assay

Tumor cells were plated at 1000 cells per well on PolyHema-coated 12-well plates in Prime-XV Tumorsphere SFM media (Irvine Scientific #91130). Cells were monitored for up to 10 days and the total number of mammospheres per well was counted. Mammospheres were defined as groups with 25 or more cells originating from a single cell.

## Quantification and statistical analysis

All data unless otherwise specified is displayed as mean ± standard deviation (SD). For comparing two groups, we employed Student's t-tests. When comparing more than two groups, we utilized ANOVA and other methods for analysis. Statistical analysis was conducted using Microsoft Excel and GraphPad to calculate p-values, with significance set at $p = <0.05$. The Kaplan-Meier curves were generated using the Kaplan-Meier plotter (https://kmplot.com/). The Cox-Mantel (log-rank) test was used to evaluate the significance of differences between two patient cohorts, assessing whether the overall survival patterns differed throughout the entire study period. The difference between the cohorts is quantified by the hazard ratio (HR), which reflects the differential decline in survival between the two groups.

## Reporting summary

Further information on research design is available in the Nature Portfolio Reporting Summary linked to this article.

## Data availability

The RAW global MS data and the identified results generated in this study have been deposited in the Japan ProteOme STandard Repository (jPOST: https://repository.jpostdb.org/) under accession code JPST002098 for jPOST and PXD041009 for ProteomeXchange (https://repository.jpostdb.org/entry/JPST002098). Raw data supporting the findings of this study are available in the Source Data files, which include uncropped blot images, raw data corresponding to each figure. Source data are provided with this paper.

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

## Acknowledgements

This work could not have been completed without the gracious contributions of the core facilities at Northwestern University including the Robert H. Lurie Comprehensive Cancer Center CTC Core, the Flow Cytometry Core, the Pathology Core, the Mouse Histology and Phenotyping Laboratory, and the Center for Comparative Medicine. We are deeply grateful to our collaborators and the Mass Spectrometry Core at Pacific Northwest National Laboratory. We would also like to thank our collaborators and all other Liu laboratory members at Northwestern University Feinberg School of Medicine. This project has been partially supported by Department of Defense grants W81XWH-16-1-0021 and W81XWH-20-1-0679 (H. Liu); NIH/NCI grants R01CA245699 (H. Liu and E.K. Ramos), R01AI167272, and R01CA298232 (H. Liu), R01GM139858 (T. Shi), and UG3CA256967 (T. Shi and H. Liu); the Lurie Cancer Center Lynn Sage Breast Cancer Research Foundation (H. Liu and N. Dashzeveg); American Cancer Society CSCC-Team-23-980420-01-CSCC (H. Liu); Chan Zuckerberg AGMT (H. Liu), a Northwestern University Pharmacology start-up grant (H. Liu); and NIH Fellowships T32 CA009560 (E.K. Ramos), T32 CA080621-15 and the Julius Kahn Fellowship (R. Taftaf), and T32GM008061 (E.J. Schuster).

## Author contributions

E.S., N.D., F.T., and H.L. conceived the idea presented here. H.L., T.S., and C-F.T. supervised experimental planning and implementation. N.D. performed an initial analysis of patient survival association with protein candidates identified from multiple datasets and breast cancer cells. E.S. drove the development of the presented storyline and data acquisition. E.S. and N.D. planned and carried out cell culture, clustering, imaging, flow cytometry/sorting, immunoprecipitation, and mass spectrometry sample preparation experiments. F. T. helped with the clustering experiments and data analyses. E.S., N.D., and F.T. planned and completed all animal work in vivo with the help of Y.J. and, later, L.E., D.P.S., and W.A.M. N.D. and G.K. carried out the EV mass spectrometry screen, including all necessary EV isolation and sample preparation. C-F.T. and R.B.K. performed mass spectrometry proteomic analyses of breast cancer cells and the co-immunoprecipitation samples, analyzed and deposited data. C.Z. and R.X. further helped analyze the mass spectrometry proteomic data of EVs. T.Z. and A.H. helped run the computational analysis across multiple cancer data sets to rank surface protein expression in breast cancer. M.C., A.S, W.J.G., L.C.P., C.R., and S.S. supported blood sample collection from breast cancer patients by enrolling patients, collecting blood work regularly for analysis, and managing patient data. E.S., N.D., Y.J., E.R., R.T., L.E-S., D.S., and V.A.C. all helped to process and analyze patient blood samples via flow cytometry. Y.Z. processed and analyzed patient blood samples via Cell-Search® with the help of S.S., T.S., R.B.K., and C.F., ran and analyzed multiple mass spectrometry analyses on breast cancer CTCs and clusters. K.P.S. aided in the analysis of TMA data. J.S. helped to analyze images of CellSearch. H.A., A.M., and W.P. helped with Western blots. D.S. performed ImageStream analysis to validate the presence of CTC clusters and immune cells. E.S. and H.L. wrote the manuscript. F.T., N.K.D., T.S., Z.T., R.B.K., L.E-S., K.P.S. and V.A.C. edited the text.

## Competing interests

Huiping Liu and Andrew D. Hoffmann are scientific co-founders and equity shareholders of ExoMira Medicine Inc., whose business is not currently related to the content of this manuscript. Massimo Cristofanilli has the following financial relationships to disclose: (1) serving as consultant for AZ, Celcuity, Menarini-Stemline, Repare Therapeutics, Olaris, Syantra, BriaCell, Datar Cancer Genomics, Biotheryx; (2) received grant or research support from AZ, Celcuity; (3) received honoraria from AZ, Merck, Iylon, Menarini-Silicon Biosystem, Datar Cancer Genomics; and (4) participated in Speaker Bureau of Pfizer. Carolina Reduzzi receives research support from Menarini Silicon Biosystems, ANGLE, Qiagen, BioRad, Tethis. The remaining authors declare no competing interests.
