## [Transparent Peer Review file · Nature Communications]

Computational ranking identifies Plexin-B2 in circulating tumor cell clustering with monocytes in breast cancer metastasis

Corresponding Author: Dr Huiping Liu

Version 0:

Reviewer comments:

Reviewer #1

(Remarks to the Author)

General Comments: The authors have made considerable effort to revise the manuscript in response to the original comments which has overall strengthened the value of their novel Rscore method and enriched their hypothesis as to the role of PLXNB2 in CTC cluster-mediated metastasis. In addition, they have improved the clarity of their figures and advanced the description of statistical methods which has improved the readability and robustness of their work. However, there are a few areas which could continue to benefit from further clarification. Moreover, to ensure accurate and clear conclusions are drawn from their work, this paper continues to require a complete editing process to be ready for publication.

Specific Comments:

1. Whilst the authors notably improved the interpretability of their Rscore method, it is still challenging to estimate the true value of their method due to two main issues. First, the new methods section notes “the individual ranks in the study include (but not limited to)”. While I appreciate this configurable method may be in place for a specific purpose (such as to incorporate information which may be present for some proteins but not others), this continues to keep the method potentially quite opaque. i.e., if one or even several other inputs which are totally unknown to the reader are responsible for high rank of PLXNB2 this is critical for the reader to understand but cannot be known from the current manuscript. More critically, the novelty and validity of this ranking is ostensibly the data science method which was used to determine the constant terms for this algorithm – i.e., was there a training set with known clinical importance used to determine constant values through an ML like XGBoost? Without knowledge of how these terms are generated it is hard to appreciate the proposed novelty of this method; however, where this paper can be accepted on the merits of its discussion of the novel identification of PLXNB2 as a protein playing a key role in breast cancer metastasis, this is not paramount.

2. The authors have in some cases used evidence which demonstrates, albeit robust, correlation to conclude a causal role for PLXNB2 in metastasis.

o In Figure 2 the authors are aiming to conclude that the only way in which a dual color metastasis can occur is through the formation of a CTC in vasculature between cells derived from both contra-lateral MF tumors which then deposits in the lung. However, I am not confident this is widely accepted. For example, a prior study in Cancer Discovery in 2016 (Maddipati et al. 2015) in a study of pancreatic polyclonal (identified by polychromic nature) metastasis concluded “As with diaphragmatic and liver metastases, polyclonal liver and lung metastases could arise via seeding of polyclonal lesions present in the circulation or by seeding of one clone followed by subsequent seeding by a second clone.” Thus, while the correlation between PLXNB2 mediated CTC clusters and lung metastasis is clear, I feel additional evidence would be needed if the authors are to draw a definitive causal link.

o Figure 5n is similarly a correlational experiment which relies on the absence of alternative explanations, and here the potential explanation of THP1 monocyte co-culture on tumor cell tumorigenicity is not addressed. Notably many works have shown that tumor cell co-culture with immune cells can alter their tumorigenic potential, perhaps most aptly Zhang et al 2021 demonstrated THP-1 macrophages enhances stemness of A549 lung adenocarcinoma with effects unrelated to CTC formation. Thus, it cannot be stated definitively that there is a casual link here without additional evidence.

Of note, the authors write in the text “Notably, at both timepoints, the presence of THP1 monocytes in heterotypic tumor cell clusters significantly increased the metastatic colonization into the lungs compared to control tumor cells alone, whereas

THP1 cells had no significant influence on PB2 KO tumor cells (Fig. 5n-o)". In contrast, their statistical analysis in 5o shows there is a statistically significant difference in metastasis in the PB2 KO + THP1 versus PB2 KO at the 4-day timepoint. This is both hard to resolve with their hypothesis and thus should be addressed. More critically, this renders the text as it currently reads somewhat misleading.

Additional Comments

- In the use of KMPlotter graphs consistent to establish correlation between PLXNB2 and prognosis, it would be more compelling to use the same operationalization of prognosis in each --- i.e., using OS for protein expression in TNBC but then DMFS for mRNA in ER(-) prompts the reader to consider if protein expression in TNBC is not correlated with an improvement in DMFS and the same for OS in ER(-) mRNA.
- The authors, in response to a prior comment, have removed the discussion of Sema7 from their data in Figure 4. However, they left comments regarding Sema7 in the manuscript (paragraph beginning on line 322)
 - o "whereas Sema4D and Sema7A were nearly undetectable (Fig. 4c)"
- The method of quantification for BLI should be made consistent. Figure 2e uses simply ROI/tumor weight in the axis which does not inform the reader what method of quantification was used; however, the caption suggests counts. In contrast, figure 2e simply states "Lung BLI", for which the method of quantification is ambiguous -- and the caption is absent for this panel. In Figure 3l-o, one quantification panel uses mean photon count while another uses simply counts. Later in Figure 5 it appropriately reports in the axis legend that bioluminescence was quantified as photons per second and then in Figure 6g as total flux, which are notably the same method but labeled differently here. First, labeling should be consistent. More critically, it is likely most appropriate that all BLI quantification use total flux (p/s), and thus be aligned with Figure 5/6, especially in cases where exposure time may vary -- and this should be reflected in the axis and caption. If there is a specific reason to vary the method of quantification from total flux, this should be addressed.
- There is no caption for Figure 1e or Figure 1f.
- The caption for Figure 2 is incorrect "o-p). Bar graphs of blood CTC clusters and cell cycle phases (p) and spontaneous lung metastasis (q) between the mice bearing WT (8 weeks) and KO tumors (10 weeks). CTCs (cluster count ratios) and cell cycle phases were measured by flow cytometry with L2T or L2G-labeled tumor cells. N=10 mice." Describes only figure 2o. Thus, there is no caption for Figure 2p.
- In Figure 6e the scale for the BLI seems to show a negative flux as well as pixelated non-focal images -- i.e., in place of areas of high density surrounded by lower density areas, the highest densities are juxtaposed to areas of very low density which suggests very low signal or an issue with measurement. Please check this data to ensure it accurately supports the conclusion of reduced metastasis and is aligned with the histopathology data in Figure S9.

Minor Technical Comments:

- The authors use different axis labels for the same method in different figures, e.g., the use of normalized cluster area in Figure 5 in contrast to average cluster area used throughout the prior figures and use variable methods to describe BLI quantification.
- Line 380 defines PMBC as macrophages and monocytes. PMBC are all mononuclear cells in blood which critically includes lymphocytes (as well as dendritic cells) but excludes PMNs.
- Figure S1 caption: "KM plot for distant metastasis-free survival (DMFS) of grade 3 breast cancer with high vs. low PLXNB2 (PB2) mRNA expression, N=836 (top) and mRNA expression of OS of breast cancer with high vs. low PB2 expression, N=2976 (data from KMPlotter). P-values were calculated using log rank test."
- Nomenclature should again be checked -- e.g., Plxnb2 KO in new supplemental figure legend should be italicized
- Check the defining of acronyms throughout, e.g.,
 - o EV acronym is undefined at first use
 - o DMFS defined at in line 158 after previously already being defined, in contrast to OS is line 157
- Line 120 -- missing a parenthetical "Extended Data Excel 1-tab 1)"
- Other general editing: Inconsistent capitalization in figure legends, inconsistent spacing between citations and final word in sentence, missing spaces between period and start of new sentence (line 498), etc.

Reviewer #3

(Remarks to the Author)

The authors have thoroughly revised the manuscript. However, some open questions with respect to the role of Plexin-B2 in cancer cell proliferation remain (also in light of published data in Gurrupu et al., 2018, Cell Death Differ 25:1259). These open questions urge for discussion in the respective discussion section of the manuscript.

Reviewer #4

(Remarks to the Author)

The revised manuscript has made significant update. Most previous comments have been addressed. Although there is still a concern on using FACS for CTC analysis, based on the significant update and effort for addressing other comments, the concern is considered minor.

Version 1:

Reviewer comments:

Reviewer #1

(Remarks to the Author)

The authors have further improved the manuscript with further revisions to address the remaining concerns. The manuscript is appropriate for publication now.

"Plexin-B2 mediates tumor cell clustering with monocyte interactions in breast cancer metastasis."

Dear Editors and Reviewers:

We are grateful for your excellent review and constructive comments on our manuscript "Plexin-B2 mediates tumor cell clustering with monocyte interactions in breast cancer metastasis." We have made substantial revisions to address all the questions thoroughly with point-to-point answers supported by new data and clarifications (in blue) and modified text (in purple).

Reviewer Comments and Responses:

Responses to Reviewer 1 comments are on pages 1-8.

Responses to Reviewer 3 comments are on page 8.

Responses to Reviewer 4 comments are on page 9.

Reviewer #1:

General Comments: The authors have made considerable effort to revise the manuscript in response to the original comments which has overall strengthened the value of their novel Rscore method and enriched their hypothesis as to the role of PLXNB2 in CTC cluster-mediated metastasis. In addition, they have improved the clarity of their figures and advanced the description of statistical methods which has improved the readability and robustness of their work. However, there are a few areas which could continue to benefit from further clarification. Moreover, to ensure accurate and clear conclusions are drawn from their work, this paper continues to require a complete editing process to be ready for publication.

We are grateful for the recognition of our efforts to revise the manuscript, clarify our figures, and strengthen the statistical rigor and scientific significance of our study. We have carefully addressed the remaining areas that required clarification and have thoroughly revised the manuscript for clarity, overall readability, as suggested. We appreciate the comments that have further helped us improve the quality and accessibility of the work. The following are point-to-point responses.

Specific Comments:

1.1 Whilst the authors notably improved the interpretability of their Rscore method, it is still challenging to estimate the true value of their method due to two main issues. First, the new methods section notes "the individual ranks in the study include (but not limited to)". While I appreciate this configurable method may be in place for a specific purpose (such as to incorporate information which may be present for some proteins but not others), this continues to keep the method potentially quite opaque. i.e., if one or even several other inputs which are totally unknown to the reader are responsible for high rank of PLXNB2 this is critical for the reader to understand but cannot be known from the current manuscript. More critically, the novelty and validity of this ranking is ostensibly the data science method which was used to determine the constant terms for this algorithm – i.e., was there a training set with known clinical importance used to determine constant values through an ML like XGBoost? Without knowledge of how these terms are generated it is hard to appreciate the proposed novelty of this method; however, where this paper can be accepted on the merits of its discussion of the novel identification of PLXNB2 as a protein playing a key role in breast cancer metastasis, this is not paramount.

Thank you for the comment regarding the RScore method and the appreciation of "the merits of its discussion of the novel identification of PLXNB2 as a protein playing a key role in breast cancer

metastasis”. We admit there are limitations of current work, i.e. possibly unknown input influencing the ranking system and acknowledge that in the “Discussion” (see below). We agree with the reviewer about the novelty of the data science method to determine the constant terms for this algorithm. Nevertheless, the current work is a pilot study with preliminary constant factors, and its ML training requires comprehensive, large dataset coupled with advanced computational methods (XGBoost etc) as well as validation with ground truth of experimentally validated ranking and clinical importance. That is our ongoing work beyond the scope of the current study due to the labor-intensive work in generating the inclusive datasets and ground truth ranking for a massive list of proteins. We are grateful for the reviewer’s understanding and admitting “this is not paramount”. Therefore, we clarified with more information in the methods and added the ongoing directions of ML training in the “Discussion” section as below.

Ongoing studies remain necessary to overcome the limitations of our study. First, the RScore approach for ranking candidate proteins is a semi-quantitative, heuristic-based method that integrates multiple modalities of biological datasets and clinical relevance. It is designed with a modifiable constant factor (weight) for each data input to provide room and flexibility for continuous training and machine learning (ML)-based determination in following studies. It will be optimized through integration with expanded dataset inputs, both known and unknown, and experimental validation of candidate rankings to enhance predictive power.

1.2 The authors have in some cases used evidence which demonstrates, albeit robust, correlation to conclude a causal role for PLXNB2 in metastasis. In Figure 2 the authors are aiming to conclude that the only way in which a dual color metastasis can occur is through the formation of a CTC in vasculature between cells derived from both contra-lateral MF tumors which then deposits in the lung. However, I am not confident this is widely accepted. For example, a prior study in Cancer Discovery in 2016 (Maddipati et al. 2015) in a study of pancreatic polyclonal (identified by polychromic nature) metastasis concluded “As with diaphragmatic and liver metastases, polyclonal liver and lung metastases could arise via seeding of polyclonal lesions present in the circulation or by seeding of one clone followed by subsequent seeding by a second clone.” Thus, while the correlation between PLXNB2 mediated CTC clusters and lung metastasis is clear, I feel additional evidence would be needed if the authors are to draw a definitive causal link.

Thank you for this insightful comment. We agree that polyclonal metastases can arise from pre-formed CTC clusters as well as sequential seeding events. However, the former is a primary contributor or more dominant than latter as demonstrated in the reviewer-cited Cancer Discovery 2015 paper by Maddipati et al (reference 52 in our manuscript). They demonstrated that the polyclonal metastases to the diaphragm (Figure 4) and lungs (Figure 6) are primarily from polyclonal CTC clusters (single CTCs injected at different times did not form polyclonal colonies at both organs). Their statements are: “Taken together, these data suggest that polyclonal peritoneal metastases develop from multiclonal aggregates shed from the primary tumor.... Consistent with our previous results, all resulting metastases were monochromatic (Fig. 6E), suggesting that multicolored metastases in this model are likely to come from seeding by polyclonal clusters of tumor cells rather than sequential rounds of seeding.” Consistently, our previous publication in Cancer Discovery 2019 (Liu X, et al) also demonstrated that in comparison to sequentially inoculated single cancer cells in different colors, the simultaneously seeded CTC clusters preferentially contribute to polyclonal metastasis to the lungs (reference 16, Figure 1K-L). Therefore, we have modified the result section description as an “association” between CTC clusters and co-colonization (see below).

Modified Result section:

To determine if PLXNB2-mediated CTC clustering was associated with increased co-colonization in metastasis, we orthotopically implanted red L2T- and green L2G-labeled *PLXNB2*⁺ WT control tumors (ConT and ConG) and *PB2*⁻ KO tumors (KOT and KOG) into separate left and right 4th mammary glands with 4 groups of combinations: (1) ConT-ConG, (2) KOT-KOG, (3) ConT-KOG, and (4) KOT-ConG (**Fig. 2i**). After 6 weeks of orthotopic tumor growth, only the mice bearing the ConT-ConG tumors in dual colors showed dual-color lung colonies whereas the counts of both single-color and dual-color metastatic colonies dramatically decreased in any of the three groups with one or two KO tumor implants (**Fig. 2j-m**). The CTC clusters were dramatically in higher frequencies in control tumor-bearing mice than the KO tumor-bearing mice (**Fig 2m**). These data are in consistency with previous demonstrations that lung co-colonization (polyclonality) is primarily contributed by CTC clusters of breast cancer (16) and pancreatic cancer (52), **albeit a small possibility of sequential seeding of polyclonal tumors.**

Cited Figure 1K-L (Liu X, et al, Cancer Discovery 2019): sequential seeding of single tumor cells within 2 hours apart depleted most of co-colonization.

[FIGURE REDACTED]

Cited Figure 1K, Cluster formation within the lung vasculature imaged ex vivo at 2 hours after tail-vein infusion of eGFP⁺ (green) and tdTomato⁺ (red) MDA-MB-231 cells at 1:1 ratio, either mixed coinfusion (0 minutes apart), or separate infusions of tdTomato⁺ cells first and then eGFP⁺ cells lagged at 5 minutes, 10 minutes, and 2 hours. Ex vivo lung fluorescence images were taken 2 hours after infusion of eGFP⁺ cells. Scale bars, 50 μ m.

L, Quantitative proportions of single-color and mixed-color clusters (lung colonies) from the four groups in K.

Cited Figure 4C-D (Maddipati et al, Cancer Discovery 2015):

[FIGURE REDACTED]

Cited Figure 4C, bar graph depicting mean percentage of total gross monochromatic (RFP or YFP only) or polychromatic (positive for both YFP and RFP) metastases between single-cell and cluster injection groups. Data pooled from n = 4 mice for each group (a total of 24 lesions were counted in the single-cell group and 50 lesions were counted in the cluster group). No lesions from the single-cell injection group were polychromatic. *, P < 0.001 by the Fisher exact test comparing multicolor metastases between single-cell and cluster injections.

D, bar graph depicting the mean number of gross metastases in the single-cell and cluster injection groups (n = 4 mice in each group).

Cited Figure 6D-E (Maddipati et al, Cancer Discovery 2015):

[FIGURE REDACTED]

Cited Figure 6C, retro-orbital injection of 458d_R and 458d_Y cells (20,000) either as mixture of single cells (top) or multicolor clusters (bottom) into NOD.SCID mice. Right, representative fluorescent images of resulting metastatic lung lesions in the two injection groups.
D, quantification of the data in C. The mean percentage of monochromatic (RFP or YFP only) or polychromatic (positive for both RFP and YFP) metastases are indicated in stacked graph format for each injection group (single cell or cluster). Data are pooled from n = 4 mice for each group (a total of 208 lesions were counted for the single-cell group and 607 lesions were counted for the cluster group). *, P < 0.001 by the Fisher exact test comparing the frequency of polychromatic metastases between single-cell and cluster injections.
E, retro-orbital injection of 458d_R and 458d_Y cells as sequential injections of single cells into NOD.SCID mice separated by 3 days. Right, representative fluorescent images of lung metastatic lesions detectable 21 days later. Table shows total metastatic counts and the percentage of monochromatic and polychromatic lesions. Data pooled from n = 4 mice. *, P < 0.001 by the Fisher exact test.

1.3 Figure 5n is similarly a correlational experiment which relies on the absence of alternative explanations, and here the potential explanation of THP1 monocyte co-culture on tumor cell tumorigenicity is not addressed. Notably many works have shown that tumor cell co-culture with immune cells can alter their tumorigenic potential, perhaps most aptly Zhang et al 2021 demonstrated THP-1 macrophages enhances stemness of A549 lung adenocarcinoma with effects unrelated to CTC formation. Thus, it cannot be stated definitively that there is a casual link here without additional evidence.

Thank you for pointing this out. We agree that THP1 cells in co-culture may influence tumor cell behavior in addition to heterotypic clustering. We have modified our conclusion accordingly and cited the paper Zhang et al. (2021).

We then compared the outcomes of experimental colonization with TNBC (MDA-MB-231) control cells and PB2 KO cells which were pre-clustered for 4 h ex vivo prior to tail vein-injections. With minimal clustering capacity, the PB2 KO cells showed a significant reduction in tumor cell dissemination to the lungs after 12 h, and the phenotype was maintained for up to four days (**Fig. 5n**), suggesting that PLXNB2 enhances dissemination and metastatic colonization, coupled with tumor clustering and independent of proliferation effects. Similarly, after 4 h pre-clustering with unlabeled THP1 monocytes, heterotypic L2G⁺ tumor cell-THP1 clusters promoted tumor cell dissemination (12 h), in a PLXNB2-dependent manner (**Fig. 5n-o**). Meanwhile, the presence of THP1 monocytes promoted the metastatic colonization of both WT and PB2 KO tumor cells within 4 days (**Fig. 5n-o**), possibly through both PLXNB2-dependent clustering and clustering-independent factors, such as stemness (69). These results demonstrated that monocytes promote heterotypic CTC clustering, early dissemination, and lung colonization.

1.4 Of note, the authors write in the text “Notably, at both timepoints, the presence of THP1 monocytes in heterotypic tumor cell clusters significantly increased the metastatic colonization into the lungs compared to control tumor cells alone, whereas THP1 cells had no significant influence on PB2 KO tumor cells (Fig. 5n-o)”. In contrast, their statistical analysis in 5o shows there is a statistically significant difference in metastasis in the PB2 KO + THP1 versus PB2 KO at the 4-day timepoint. This is both hard to resolve with their hypothesis and thus should be addressed. More critically, this renders the text as it currently reads somewhat misleading.

Thank you for your comment. We have revised the text to better align with the results and to avoid any potential misinterpretation.

Similarly, after 4 h pre-clustering with unlabeled THP1 monocytes, heterotypic L2G⁺ tumor cell-THP1 clusters promoted tumor cell dissemination (12 h), in a PLXNB2-dependent manner (Fig. 5n-o). Meanwhile, the presence of THP1 monocytes promoted the metastatic colonization of both WT and PB2 KO tumor cells within 4 days (Fig. 5n-o), possibly through both PLXNB2-dependent clustering and clustering-independent factors, such as stemness (69).

1.5 In the use of KMPlotter graphs consistent to establish correlation between PLXNB2 and prognosis, it would be more compelling to use the same operationalization of prognosis in each --- i.e., using OS for protein expression in TNBC but then DMFS for mRNA in ER(-) prompts the reader to consider if protein expression in TNBC is not correlated with an improvement in DMFS and the same for OS in ER(-) mRNA.

Thank you for this comment. We agree that using a consistent endpoint across analyses would strengthen the interpretation of the correlation between PLXNB2 expression and prognosis. However, due to limitations in publicly available datasets, particularly the relatively small number of TNBC patients with corresponding clinical and expression data, we included all breast cancer patients for the OS analysis based on both protein and mRNA levels of PLXNB2 (Fig 1c and Extended Fig S1f). For the DMFS analysis, protein-level data were not available. To address this limitation and improve consistency, we have now generated and included a new KM plot analyzing DMFS based on mRNA expression of *PLXNB2* in all breast cancer patients (now included in Extended Fig S1f). Additionally, in response to the reviewer’s concern regarding the ER(-) group, we previously showed in Fig 1d that *PLXNB2* mRNA expression is associated with DMFS in ER-negative breast cancer patients. To further address this point, we have now generated an additional KM plot analyzing OS based on *PLXNB2* mRNA levels specifically in ER-negative patients, which is included in Extended Fig S1h. These new KM plots improve clarity and strengthen the evidence for a clinical association between PLXNB2 expression (at both the protein and mRNA levels) and prognosis in breast cancer patients, highlighting the potential importance of PLXNB2 in breast cancer progression.

Figure 1. c) KM plot for OS of patients with all breast cancer in the Tang_2018 data set (N=108) via Kaplan-Meier plotter, separated by the best cut-off value of PLXNB2 protein expression (4) in primary tumors to define high vs. low within the expression range (0-11). P values were calculated via the Cox-Mantel (log-rank) test.

d) KM plot for DMFS of patients with ER breast cancer, divided by median cut-off of high vs. low PLXNB2 mRNA expression using data from GEO, EGA, and TCGA, N=218. P-values were calculated using a log-rank test.

1f) KM plot for distant metastasis-free survival (DMFS) of grade 3 breast cancer (N=836) with high vs. low PLXNB2 mRNA expression (data from KMPlotter), OS of all breast cancer (N=2976) with high vs. low PLXNB2 mRNA expression (data from KMPlotter), DMFS of all breast cancer (N=198) with high vs. low PLXNB2 mRNA expression using data from GEO (GSE7390), separated by best cut-off values. P-values were calculated using log rank test.

1h) KM plot for OS of ER- breast cancer with high vs. low PLXNB2 mRNA expression using data from GEO (GSE58812), N=107, separated by best cut-off values. P-values were calculated using a log-rank test.

1.6 The authors, in response to a prior comment, have removed the discussion of Sema7 from their data in Figure 4. However, they left comments regarding Sema7 in the manuscript (paragraph beginning on line 322) o “whereas Sema4D and Sema7A were nearly undetectable (Fig. 4c)”

Thanks for pointing it out. We have revised the manuscript accordingly.

1.7 The method of quantification for BLI should be made consistent. Figure 2e uses simply ROI/tumor weight in the axis which does not inform the reader what method of quantification was used; however, the caption suggests counts. In contrast, figure 2e simply states “Lung BLI”, for which the method of quantification is ambiguous -- and the caption is absent for this panel. In Figure 3l-o, one quantification panel uses mean photon count while another uses simply counts. Later in Figure 5 it appropriately reports in the axis legend that bioluminescence was quantified as photons per second and then in Figure 6g as total flux, which are notably the same method but labeled differently here. First, labeling should be consistent. More critically, it is likely most appropriate that all BLI quantification use total flux (p/s), and thus be aligned with Figure 5/6, especially in cases where exposure time may vary -- and this should be reflected in the axis and caption. If there is a specific reason to vary the method of quantification from total flux, this should be addressed.

Thank you for the comment. We have modified the labels to be consistent and added more details in the methods.

Figure 2e, 2q: BLI Total Flux/ tumor weight (p/s/g, $\times 10^5$)

Figures 3m, 3o: BLI Total Flux (p/s)

Figure 5o: BLI Total Flux (p/s)

Figure 6g: BLI Total Flux (p/s)

Extended Figure S4d: BLI Total Flux (p/s)

Methods section:

The bioluminescence imaging (BLI) signals for regions of interest were measured as total flux (photons/second, p/s). When applicable, the tumor growth signals were normalized and presented as fold change in comparison to that of Day 0. To assess spontaneous lung metastasis, the BLI signals (total flux, p/s) were normalized by tumor weight to minimize the effects of tumor burden.

1.8 There is no caption for Figure 1e or Figure 1f.

Thanks for pointing it out. We have added the captions for both Figure 1e and 1f.

1.9 The caption for Figure 2 is incorrect “o-p). Bar graphs of blood CTC clusters and cell cycle phases (p) and spontaneous lung metastasis (q) between the mice bearing WT (8 weeks) and KO tumors (10 weeks). CTCs (cluster count ratios) and cell cycle phases were measured by flow cytometry with L2T or L2G-labeled tumor cells. N=10 mice.” Describes only figure 2o. Thus, there is no caption for Figure 2p.

We have revised the caption accordingly and added “(o), (p), and (q)” after “blood CTC clusters”, “cell cycle phases” and “spontaneous lung metastasis”, respectively.

1.10 In Figure 6e the scale for the BLI seems to show a negative flux as well as pixelated non-focal images – i.e., in place of areas of high density surrounded by lower density areas, the highest densities are juxtaposed to areas of very low density which suggests very low signal or an issue with measurement. Please check this data to ensure it accurately supports the conclusion of reduced metastasis and is aligned with the histopathology data in Figure S9.

Thank you for the comment. The BLI scale in Figure 6e uses dashes (-) to indicate the position of numerical values, but not negative value of the flux. To avoid confusion, we have moved the dash for better clarity. The H&E staining shown in Figure S9e corresponds to the same *ex vivo* lung samples used for BLI imaging in Figure 6e. We have updated the figure to ensure consistency between the BLI and histopathology data.

1.11 The authors use different axis labels for the same method in different figures, e.g., the use of normalized cluster area in Figure 5 in contrast to average cluster area used throughout the prior figures and use variable methods to describe BLI quantification.

Thanks for pointing it out. We have carefully revised the y axis labels as “Average cluster area (μm^2 , $\times 10^3$)” to be consistent.

1.12 Line 380 defines PMBC as macrophages and monocytes. PMBC are all mononuclear cells in blood which critically includes lymphocytes (as well as dendritic cells) but excludes PMNs.

Thank you for pointing out that. We have revised accordingly.

1.13 Figure S1 caption: “KM plot for distant metastasis-free survival (DMFS) of grade 3 breast cancer with high vs. low PLXNB2 (PB2) mRNA expression, N=836 (top) and mRNA expression of OS of breast cancer with high vs. low PB2 expression, N=2976 (data from KMPlotter). P-values were calculated using log rank test.”

Thank you for pointing out that. We have revised accordingly.

1.14 Nomenclature should again be checked – eg.g., Plxnb2 KO in new supplemental figure legend should be italicized

We have carefully reviewed the manuscript and figure legends to ensure consistent and correct gene/protein nomenclature.

1.15 Check the defining of acronyms throughout, e.g.,

o EV acronym is undefined at first use

We have added the definition of EV at first use.

o DMFS defined at in line 158 after previously already being defined, in contrast to OS is line 157

We have removed the repeated definition of DMFS.

• Line 120 – missing a parenthetical “Extended Data Excel 1-tab 1)”

Parenthetical added.

• Other general editing: Inconsistent capitalization in figure legends, inconsistent spacing between citations and final word in sentence, missing spaces between period and start of new sentence (line 498), etc.

We have edited accordingly.

Reviewer #3:

The authors have thoroughly revised the manuscript. However, some open questions with respect to the role of Plexin-B2 in cancer cell proliferation remain (also in light of published data in Gurrupu et al., 2018, Cell Death Differ 25:1259). These open questions urge for discussion in the respective discussion section of the manuscript.

We appreciate the valuable comment and agree that the role of Plexin-B2 in cancer cell proliferation remains important. We have updated the Discussion section highlighting the additional role of Plexin-B2 in proliferation with the relevant citation.

PLXNB2 is identified as a driver that promotes both homotypic and heterotypic CTC clustering through its interactions with Semaphorin ligands SEMA4C on tumor cells and SEMA4A on monocytes, in addition to PLXNB2 functions in proliferation and other known functions (43-50). Gurrupu et al. demonstrated that PLXNB2 promotes breast cancer cell proliferation through the

RhoA and MAPK signaling pathways (69), which appears to be context-dependent and influenced by the expression levels of co-receptors such as MET and ErbB2.

Reviewer #4:

The revised manuscript has made significant update. Most previous comments have been addressed. Although there is still a concern on using FACS for CTC analysis, based on the significant update and effort for addressing other comments, the concern is considered minor.

Thank you for acknowledging the significant updates in the revised manuscript. We agree with your comment regarding the use of FACS for CTC analysis. We have added a discussion section that elaborates on the limitations of using FACS for CTC analysis

Second, the use of fluorescence-activated cell sorting (FACS) for analyses of CTCs and CTC clusters requires cross validation via other methods, including but not limited to multiplexing fluorescence imaging and/or single-cell sequencing of enriched CTCs from fixed or unfixed blood cells *ex vivo* (i.e., CellSearch, Parsortix, ImageStream, CellView, etc) as well as intra-vascular CTCs in tissue sections in situ (no blood processing).